# Nuclear receptor corepressor 1 controls regulatory T cell subset differentiation and effector function

**Valentina Stolz[1†], Rafael de Freitas e Silva[1†], Ramona Rica[1], Ci Zhu[1], Teresa Preglej[1], Patricia Hamminger[1], Daniela Hainberger[1], Marlis Alteneder[1], Lena Müller[1], Monika Waldherr[1], Darina Waltenberger[1], Anastasiya Hladik[2], Benedikt Agerer[3], Michael Schuster[3], Tobias Frey[4], Thomas Krausgruber[3,5], Sylvia Knapp[2], Clarissa Campbell[3], Klaus Schmetterer[4], Michael Trauner[6], Andreas Bergthaler[3,7], Christoph Bock[3,5], Nicole Boucheron[1], Wilfried Ellmeier[1]***

[1]Medical University of Vienna, Center for Pathophysiology, Infectiology and Immunology, Institute of Immunology, Vienna, Austria; [2]Medical University of Vienna, Vienna, Department of Medicine I, Laboratory of Infection Biology, Vienna, Austria; [3]CeMM Research Centre for Molecular Medicine of the Austrian Academy of Sciences, Vienna, Austria; [4]Medical University of Vienna, Department of Laboratory Medicine, Vienna, Austria; [5]Medical University of Vienna, Center for Medical Statistics, Informatics, and Intelligent Systems, Institute of Artificial Intelligence, Vienna, Austria; [6]Medical University of Vienna, Department of Internal Medicine III, Division of Gastroenterology and Hepatology, Hans Popper Laboratory of Molecular Hepatology, Vienna, Austria; [7]Medical University of Vienna, Vienna, Center for Pathophysiology, Infectiology and Immunology, Institute for Hygiene and Applied Immunology, Vienna, Austria

**\*For correspondence:** wilfried.ellmeier@meduniwien.ac.at

†These authors contributed equally to this work

**Competing interest:** The authors declare that no competing interests exist.

**Abstract** FOXP3+ regulatory T cells (Treg cells) are key for immune homeostasis. Here, we reveal that nuclear receptor corepressor 1 (NCOR1) controls naïve and effector Treg cell states. Upon NCOR1 deletion in T cells, effector Treg cell frequencies were elevated in mice and in in vitro-generated human Treg cells. NCOR1-deficient Treg cells failed to protect mice from severe weight loss and intestinal inflammation associated with CD4+ T cell transfer colitis, indicating impaired suppressive function. NCOR1 controls the transcriptional integrity of Treg cells, since effector gene signatures were already upregulated in naïve NCOR1-deficient Treg cells while effector NCOR1-deficient Treg cells failed to repress genes associated with naïve Treg cells. Moreover, genes related to cholesterol homeostasis including targets of liver X receptor (LXR) were dysregulated in NCOR1-deficient Treg cells. However, genetic ablation of LXRβ in T cells did not revert the effects of NCOR1 deficiency, indicating that NCOR1 controls naïve and effector Treg cell subset composition independent from its ability to repress LXRβ-induced gene expression. Thus, our study reveals that NCOR1 maintains naïve and effector Treg cell states via regulating their transcriptional integrity. We also reveal a critical role for this epigenetic regulator in supporting the suppressive functions of Treg cells in vivo.

## Editor's evaluation

This study presents important findings on the role of the transcriptional adaptor protein NCOR1 for mouse and human regulatory T (Treg) cell differentiation. The study shows that the LXRbeta – NCOR1 axis restricts the terminal differentiation of Treg cells into effector Tregs. It also shows that,

in addition to an impact on effector Treg differentiation, loss of NCOR1 leads to impaired suppressive function of Treg cells. The results are convincing and will contribute to our understanding of Treg cell differentiation and function.

## Introduction

Treg cells expressing the transcription factor Forkhead box protein P3 (FOXP3) play a crucial role in mediating immune tolerance to self-antigens, regulating the interaction between the host and its commensal flora, and promoting tissue repair. Deficiency or aberrant function of Treg cells triggers autoimmunity and inflammation (*Sakaguchi et al., 2020*; *Sumida et al., 2024*). Therefore, the development of these cells, as well as their effector functions, must be tightly regulated (*Josefowicz et al., 2012*; *Li and Zheng, 2015*; *Kitagawa and Sakaguchi, 2017*; *Savage et al., 2020*). The transcription factor FOXP3 is required for the generation, maintenance, and suppressive function of Treg cells, and loss of FOXP3 leads to fatal systemic autoimmunity in mice and humans (*Hori et al., 2003*; *Fontenot et al., 2003*; *Khattri et al., 2003*). Upon stimulation, naïve Treg cells (also known as resting Treg cells) undergo differentiation into effector Treg cells (also known as activated Treg cells) thereby acquiring distinct phenotypes and effector molecules expression (*Cretney et al., 2013*; *Levine et al., 2014*; *Contreras-Castillo et al., 2024*). Effector Treg cells exhibit enhanced suppressive function compared to naïve Treg cells and maintain immune tolerance and homeostasis (*Liston and Gray, 2014*; *Buszko and Shevach, 2020*). The conversion of naïve Treg cells into effector Treg cells is accompanied by metabolic reprogramming including relatively higher engagement of aerobic glycolysis over oxidative phosphorylation. This cellular change is necessary to support the function of effector Treg cells and to supply the biosynthetic material needed for their proliferation and expansion (*Shi and Chi, 2019*; *Carbone et al., 2024*). However, the transcriptional pathways involved in effector Treg cell generation remain poorly understood.

The nuclear receptor co-repressor 1 (NCOR1) is a transcriptional regulator that bridges chromatin-modifying enzymes and transcription factors. NCOR1 mediates transcriptional repression of nuclear receptors (NRs) such as thyroid hormone (TR), retinoic acid receptors (RAR), and liver X receptor (LXR) in the absence of their ligands, via its interaction with members of the HDAC family, in particular, HDAC3. NCOR1 is part of larger multi-subunit complexes and also interacts with several other transcription factors unrelated to NRs (*Hörlein et al., 1995*; *Müller et al., 2018*). We and others have recently identified NCOR1 as an important regulator of T cell development, as T cell-specific deletion of NCOR1 resulted in impaired survival of positively selected NCOR1-null TCRβ$^{hi}$CD69$^{+/-}$ thymocytes. Furthermore, NCOR1 deficiency also leads to reduced numbers of peripheral T cells including Treg cells (*Wang et al., 2017*; *Müller et al., 2017*). In addition, it was shown that NCOR1 represses the pro-apoptotic factor BIM in double-positive thymocytes post-signaling, thereby affecting thymocyte survival during positive selection (*Wang et al., 2017*). These data indicate that NCOR1 is essential for positive and negative selection during T cell development, as well as for the generation of the peripheral T cell pool. Additionally, NCOR1 was shown to regulate the transcriptional programs and effector functions of activated Th1 and Th17 cells (*Hainberger et al., 2020*). However, whether NCOR1 controls the function of immunosuppressive Treg cells has yet to be investigated.

Here, we employed conditional gene targeting approaches to study the role of NCOR1 in Treg cells. NCOR1 was deleted either in all peripheral T cells using *Ncor1*$^{f/f}$ mice crossed with *Cd4*-Cre mice (*Ncor1*$^{f/f}$*Cd4*-Cre, designated as NCOR1-cKO) or selectively in Treg cells using *Ncor1*$^{f/f}$ mice crossed with *Foxp3*-YFP-Cre mice (*Ncor1*$^{f/f}$*Foxp3*-YFP-Cre; designated as NCOR1-cKO$^{Foxp3}$). Using these mouse models, we analyzed Treg cell generation, Treg cell transcriptomes, and Treg cell function in vitro as well as in adoptive CD4$^+$ T cell transfer colitis. Moreover, we assessed whether NCOR1 is essential for human Treg cell differentiation in vitro using CRISPR-Cas9 mediated knockout approaches. Finally, since a pathway analysis indicated alterations in LXR/RXR activation pathways, we generated T cell-specific NCOR1 and LXRβ double-deficient mice to address whether NCOR1 requires LXRβ to repress effector Treg cell differentiation. Using these experimental strategies, we identified NCOR1 as a key regulatory molecule in mice and humans that controls the ratio of naïve to effector Treg cells by ensuring transcriptional integrity of Treg cell states, revealed that NCOR1 restrains effector Treg cell generation independently of LXRβ, and uncovered a positive role for NCOR1 in controlling the suppressive function of Treg cells in vivo.

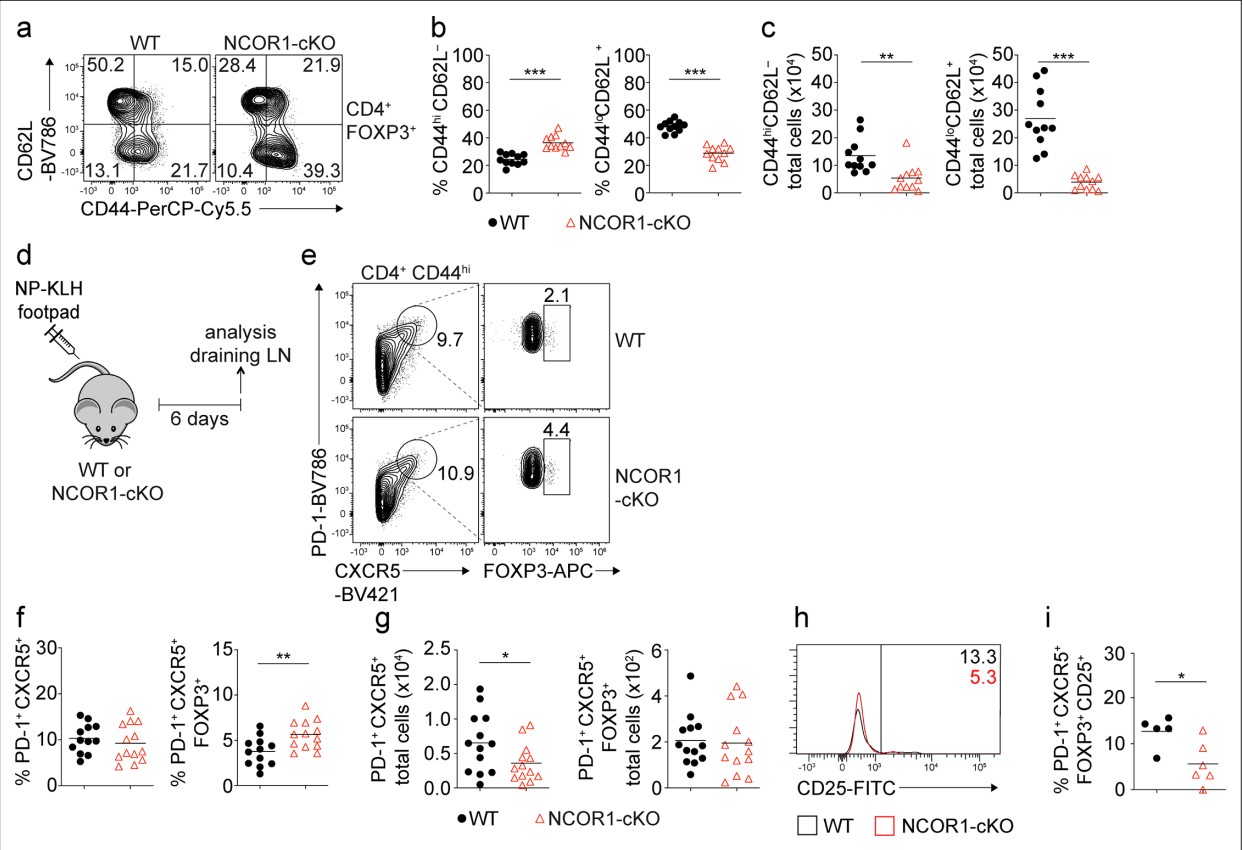

**Figure 1.** Loss of NCOR1 leads to a relative increase in CD44hiCD62L− effector Treg cells. (**a**) Flow cytometric analysis of splenocytes isolated from wild-type (WT) and NCOR1-cKO mice showing CD44 and CD62L expression in CD4+FOXP3+ cells at steady-state. (**b**) Diagrams showing the percentage of CD44hiCD62L− (left) and CD44loCD62L+ (right) CD4+FOXP3+ cells of all mice analyzed as described in (**a**). (**c**) Total cell numbers of CD44hiCD62L− (left) and CD44loCD62L+ (right) CD4+FOXP3+ cells of all mice analyzed are described in (**a**). (**d**) Experimental immunization strategy: mice were injected s.c. with nitrophenol keyhole limpet hemocyanin (NP-KLH) and draining lymph nodes (LNs) analyzed six days later. (**e**) Flow cytometric analysis of cells isolated from draining LN of NP-KLH-immunized WT and NCOR1-cKO mice showing the expression of PD-1, CXCR5, and FOXP3 on CD4+CD44hi cells. (**f**) Diagrams showing the percentage of T follicular helper (Tfh) cells (CD4+CD44hiPD1+) (left) and T follicular regulatory (Tfr) cells (CD4+CD44hiPD1+CXCR5+FOXP3+) (right) of all mice analyzed as described in (**d,e**). (**g**) Total cell numbers of Tfh cells (CD4+CD44hiPD1+CXCR5+) (left) and Tfr cells (CD4+CD44hiPD1+CXCR5+FOXP3+) (right) of all mice analyzed as described in (**d,e**). (**h**) Flow cytometric analysis of cells isolated from draining LN of NP-KLH-immunized WT and NCOR1-cKO mice showing the expression CD25 in CD4+CD44hiPD1+CXCR5+FOXP3+ Tfr cells. (**i**) Percentage of CD25 expressing CD4+CD44hiPD1+CXCR5+FOXP3+ Tfr cells of all mice analyzed as described in (**h**). (**a,e,h**) Numbers indicate the percentages of cells in the respective quadrants or gates. (**a–c**) Cells were pre-gated on CD4 and FOXP3. (**b,c,f,g,i**) Each symbol indicates one mouse. Horizontal bars indicate the mean. *p<0.05, **p<0.01, and ***p<0.001; Unpaired two-tailed Student's t-test. Data are representative (**a,e,h**) or show the summary (**b,c,f,g,i**) of at least eleven (**b,c**), twelve (**f,g**) or five (**i**) mice that were analyzed in at least two independent experiments.

The online version of this article includes the following figure supplement(s) for figure 1:

**Figure supplement 1.** Effector markers are upregulated in NCOR1-deficient Treg cells.

**Figure supplement 2.** Characterization of FOXP3+ Treg cells isolated from NCOR1-cKO mice.

**Figure supplement 3.** Gating strategies for flow cytometric analysis.

## Results

### Increased frequencies of effector Treg cells in NCOR1-cKO mice

To investigate whether NCOR1 activity is important for FOXP3+ regulatory T cell subset differentiation and function, we characterized Treg cells in *Ncor1*f/f mice crossed with *Cd4*-Cre mice (NCOR1-cKO). Similar to conventional T cells, Treg cells can be subdivided based on CD44 and CD62L expression into naïve (CD44loCD62L+) and activated/effector (CD44hiCD62L−) populations (*Levine et al., 2014*). We observed an increase in the proportion of CD44hiCD62L− effector Treg cells and a corresponding decrease in CD44loCD62L+ naïve Treg cells in the spleen (*Figure 1a and b*), Lymph node (LN) and

mesenteric LN (mLN) (*Figure 1—figure supplement 1a and b*) in the absence of NCOR1. The relative increase in effector Treg cells in NCOR1-cKO mice also correlated with an elevated expression of KLRG1 and CD69 in Treg cells, two markers characteristic of activated/effector Treg cells (*Yu et al., 2018*; *Dias et al., 2017*; *Cortés et al., 2014*; *Cheng et al., 2012*), while the proportion of CD25[+] cells within the FOXP3[+] Treg cell population was decreased. Moreover, we detected higher percentages of ICOS-expressing Treg cells in NCOR1-cKO mice, and NCOR1-deficient Treg cells expressed elevated levels of GITR in LN and mLN. There was no significant difference in the expression of CTLA4 between WT and NCOR1-cKO Treg cells (*Figure 1—figure supplement 1c and d*). However, the total numbers of naïve and effector Treg cells were reduced in NCOR1-cKO mice (*Figure 1c*). This is due to the overall reduction in Treg cell numbers, caused by the drop in total CD4[+] T cells as well as the relative reduction of Treg cells within the CD4[+] T cell population in NCOR1-cKO mice, as previously reported (*Müller et al., 2017*; *Hainberger et al., 2020*). We also tested whether alterations in the proliferation and/or survival of NCOR1-cKO Treg cells contribute to the relative reduction in the frequency of Treg cells within the peripheral CD4[+] T cell population. The percentage of Treg cells expressing Ki67, a marker associated with cell proliferation, as well as the expression levels of Ki67 (gMFI) in FOXP3[+] cells, was similar between splenic WT and NCOR1-cKO Treg cells (*Figure 1—figure supplement 2a and b*). Similarly, there was no difference in the viability of WT and NCOR1-cKO Treg cells (see timepoint 0 hr; *Figure 1—figure supplement 2c*). These data suggest that Treg cell proliferation and survival at a steady state are not controlled by NCOR1. Of note, there was no increase in CD44[hi]CD62L[−] effector cell subsets within the FOXP3[−] CD4[+] T cell population, in fact, effector subsets were slightly reduced (*Figure 1—figure supplement 2d*). Taken together, the flow cytometry analysis indicates that the deletion of NCOR1 in T cells changes the ratio of naïve to effector Treg cells at a steady state.

We next investigated whether loss of NCOR1 affects Treg cell differentiation in the thymus. Thymic-derived Treg cells develop from CD4SP CD25[+] progenitor cells which upregulate FOXP3 expression (*Lio and Hsieh, 2008*; *Burchill et al., 2008*), although FOXP3 can also be induced in CD4SP cells before CD25 expression (*Tai et al., 2013*). In the NCOR1-cKO CD4SP population, both CD25[+]FOXP3[+] and CD25[−]FOXP3[+] subsets were reduced in comparison to WT CD4SP cells (*Figure 1—figure supplement 2e and f*). Furthermore, there was an increase in CD44[hi]CD62L[−] cells within both the CD25[−] and CD25[+] fraction of the FOXP3[+] CD4SP population in NCOR1-cKO mice (*Figure 1—figure supplement 2g*). These data indicate that the relative reduction of Treg cells within the CD4[+] T cell population as well as the change in the relative abundance of naïve and effector Treg cells in NCOR1-cKO mice is (in part) already established during the generation of Treg cells in the thymus.

## Increased effector Treg cell subsets after immunization in NCOR1-cKO mice

Treg cells acquire specialized functions in the periphery. T follicular regulatory (Tfr) cells represent a subset of peripheral effector Treg cells that are induced during an immune response. Tfr cells are defined as CXCR5[+]PD1[+]CD44[hi] CD4[+] T cells that express FOXP3 (*Wollenberg et al., 2011*; *Linterman et al., 2011*; *Chung et al., 2011*). To test whether the generation of Tfr cells is enhanced in vivo during an immune response in the absence of NCOR1, WT, and NCOR1-cKO mice were immunized subcutaneously (s.c.) with NP-KLH (nitrophenol keyhole limpet hemocyanin) mixed with alum. T cell subsets were analyzed in the draining LNs (dLNs) 6 days later (*Figure 1d*). After immunization, similar percentages of CXCR5[+]PD1[+]CD44[hi] T follicular helper (Tfh) cells were induced within the CD4[+] T cell population in WT and NCOR1-cKO mice (*Figure 1e and f*). In contrast, the frequency of Tfr cells was higher in NCOR1-cKO mice (*Figure 1e and f*), which is consistent with our observation that effector Treg cells were increased within the total Treg cell population at a steady state. The total number of CXCR5[+]PD1[+]CD44[hi] Tfh cells was slightly reduced in the absence of NCOR1, while Tfr cell numbers were similar between WT and NCOR1-cKO mice (*Figure 1g*). Furthermore, we detected decreased expression of CD25 in CXCR5[+]PD1[+]CD44[hi]FOXP3[+] Tfr cells isolated from NCOR1-cKO mice compared to WT mice (*Figure 1h and i*), in line with the observation that the most mature Tfr cells in the germinal center downregulate CD25 (*Wing et al., 2017*). Together, this data suggests that NCOR1 restrains the generation of activated/effector Treg cells not only at a steady state but also upon immunization.

To study whether NCOR1 controls the generation of activated Treg cells by modulating events induced by TCR stimulation, we next analyzed proximal TCR signaling pathways in WT and NCOR1-cKO

Treg cells ex vivo. We assessed the expression of phospho-ERK $^{T202/Y204}$, phospho-AKT$^{S473,}$ and phospho-S6$^{S240/244}$ in FOXP3$^+$ Treg cells cultured in the presence of anti-CD3/anti-CD28 for 10 min, 2 hr, and 24 hr (*Figure 1—figure supplement 2h*). Despite a trend for higher levels of phospho-ERK$^{T202/Y204}$, phospho-AKT$^{S473}$ and phospho-S6$^{S240/244}$ in NCOR1-cKO Treg cells at several early time points, we only observed a significant difference for phospho-AKT$^{S473}$ at the 2 hr timepoint, which might indicate altered mTORC2 activation (*Sarbassov et al., 2005*; *Szwed et al., 2021*). Next, we assessed whether reduced viability upon activation could contribute to the altered effector Treg cell population in NCOR1-cKO by culturing splenocytes in the presence of anti-CD3/anti-CD28 for up to 72 hr. While there was no significant difference in the viability of WT and NCOR1-cKO Treg cells after 4 or 24 hr of activation, we observed a decrease in the percentage of viable NCOR1-cKO Treg cells after 72 hr (*Figure 1—figure supplement 2c*). Overall, these data suggest that NCOR1-cKO Treg cells might display a slightly higher transient degree of activation after TCR triggering but their viability declines a few days later.

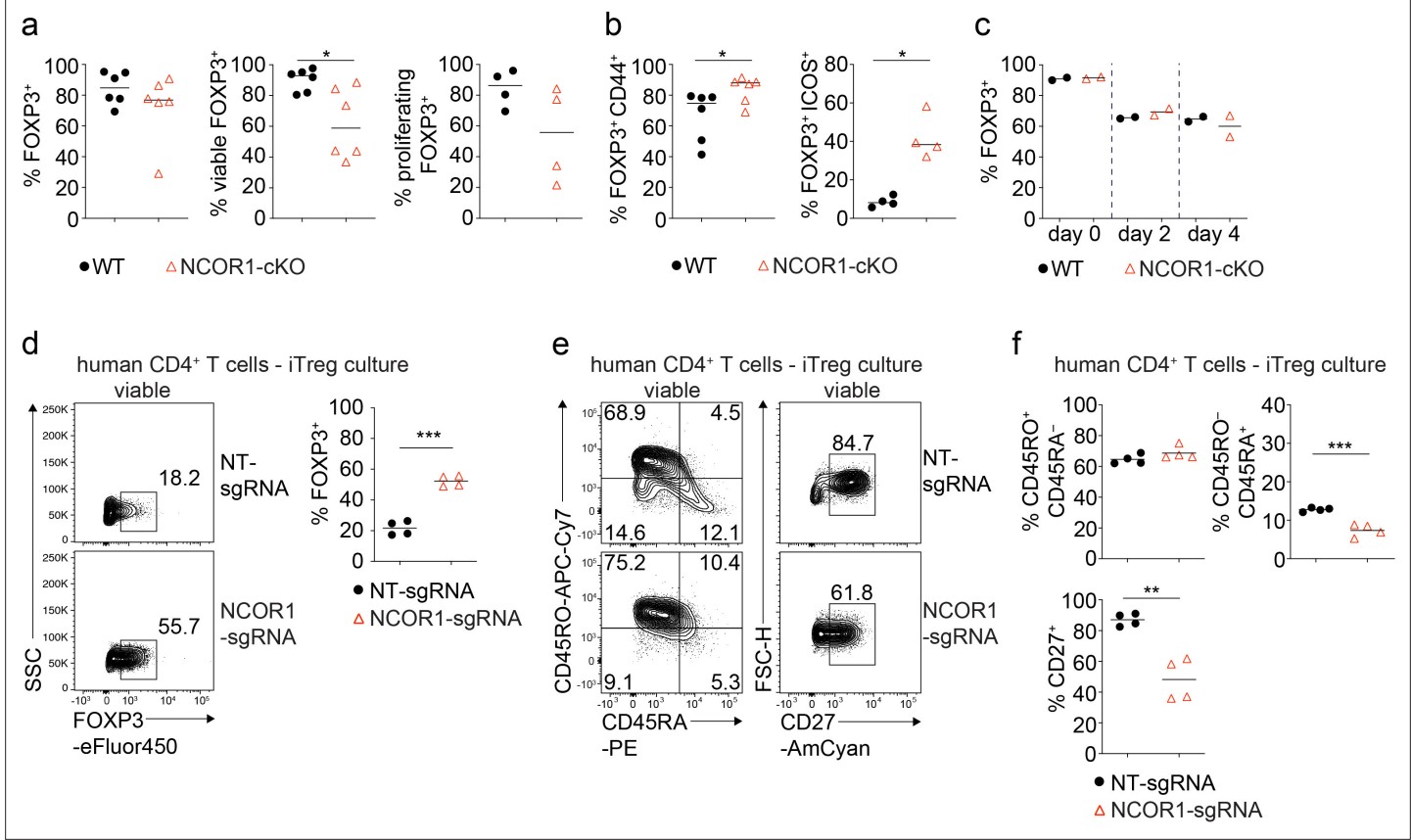

**Figure 2.** NCOR1 controls effector features in murine and human iTreg cells. (**a,b**) Naïve wild-type (WT) and NCOR1-cKO CD4$^+$ T cells were activated with anti-CD3/anti-CD28 for 3 days in the presence of TGFβ and IL2. (**a**) Diagrams show the summary of the percentages of WT and NCOR1-cKO FOXP3$^+$ (left), of viable FOXP3$^+$ (middle) and of proliferating FOXP3$^+$ (right) iTreg cells at day 3. (**b**) Summaries of the percentages of CD44$^{hi}$ FOXP3$^+$ (left) and ICOS$^+$ FOXP3$^+$ (right) cells. (**c**) Maintenance of FOXP3 expression in iTreg cells differentiated from WT and NCOR1-cKO naïve CD4$^+$ T cells over the course of 3 days upon restimulation with anti-CD3/anti-CD28 in the absence of TGFβ. (**d**) Flow cytometric analysis showing FOXP3 expression in human CD4$^+$ T cells cultured under iTreg conditions after CRISPR-Cas9-mediated knockout of NCOR1 (NCOR1-sgRNA) or in non-targeting control samples (NT-sgRNA). Diagram at the right shows the summary of all experiments. (**e**) Flow cytometric analysis showing CD45RA, CD45RO, and CD27 expression on human iTreg cells after CRISPR-Cas9-mediated NCOR1 knockout as described in (**d**). (**f**) Summaries of the experiments as described in (**e**). (**d,e**) Cells were pre-gated on the total viable cell population. (**a,b,c**) Each symbol indicates one mouse. Horizontal bars indicate the mean. *p<0.05, **p<0.01, and ***p<0.001. (**a,b,c**) Unpaired two-tailed Student's t-test. Data show a summary (**a,b,c**) of two to six mice that were analyzed in one to three experiments. (**d,f**) Each symbol indicates one sample. Horizontal bars indicate the mean. *p<0.05, **p<0.01, and ***p<0.001; unpaired two-tailed Student's t-test (**d,f**). Data are representative (**d,e**) or show the summary (**d,f**) of CD4$^+$ T cells from four individual healthy donors which were analyzed in one experiment.

## NCOR1 controls effector features in murine and human iTreg cells

To test whether NCOR1 deletion affects the differentiation or effector features of in vitro generated Treg cells (iTreg cells), we activated naïve WT and NCOR1-cKO CD4+ T cells with anti-CD3/anti-CD28 for three days in the presence of IL-2 and TGFβ. There was no difference in FOXP3 expression between WT and NCOR1-deficient iTreg cells (*Figure 2a*). However, NCOR1-cKO iTreg cells exhibited reduced viability compared to WT iTreg cells, and CFSE-labeling experiments indicated a tendedcy of reduced proliferation of NCOR1-cKO iTreg cells (*Figure 2a*). In contrast, CD44 and ICOS were upregulated in NCOR1-cKO iTreg cells compared to WT iTreg cells (*Figure 2b*), suggesting enhanced effector features in the absence of NCOR1.

iTreg cells are known to downregulate FOXP3 upon restimulation with anti-CD3/anti-CD28 in the absence of TGFβ (*Floess et al., 2007*). To test whether NCOR1 is important for the maintenance of FOXP3 expression, we restimulated WT and NCOR1-cKO iTreg cells under these conditions for 4 days. There was no difference in the downregulation of FOXP3 between WT and NCOR1-cKO Treg cells, indicating that NCOR1 function is dispensable for the maintenance of FOXP3 expression in iTreg cells (*Figure 2c*).

To investigate whether loss of NCOR1 alters the ratio of naïve to effector cells in human Treg cells, we deleted NCOR1 in human CD4+ T cells cultured in Treg-inducing conditions using CRISPR-Cas9 mediated knockout approaches. Deletion of NCOR1 led to an increase in the fraction of human CD4+ T cells expressing high levels of FOXP3 (*Figure 2d*). We also analyzed the expression of CD45RA and CD45RO to distinguish naïve (CD45RA+CD45RO−) from effector (CD45RA−CD45RO+) cells (*Terry et al., 1988*; *Sanders et al., 1988*). There was a slight reduction of NCOR1-deficient human CD4+ T

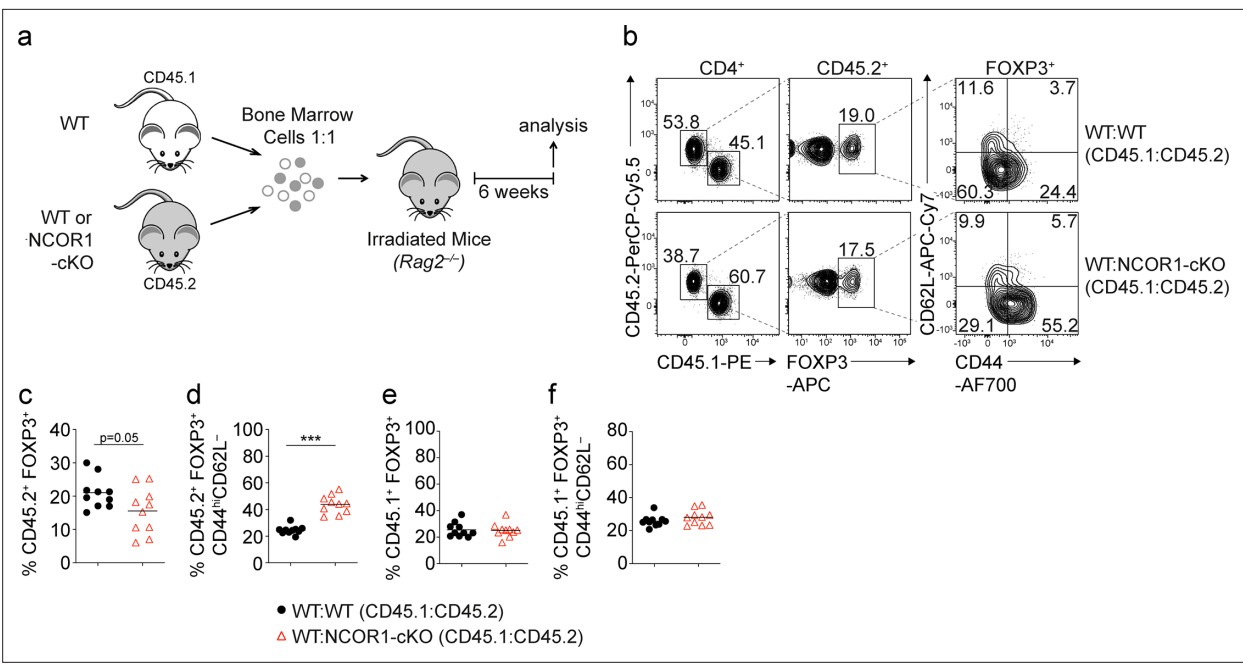

**Figure 3.** NCOR1 regulates the ratio of naive to effector Treg cells in a Treg cell-intrinsic manner. (**a**) Experimental strategy for generating bone marrow (BM) chimeric mice. (**b**) Flow cytometric analysis showing the distribution of CD45.1+ and CD45.2+ cells as well as the expression of FOXP3, CD44 and CD62L in recipient mice injected with a mix of either WT:WT or WT:NCOR1-cKO BM cells. (**c–f**) Summary of all experiments as described in (**b**) showing the percentages of FOXP3+ cells (**c,e**) and CD44hiCD62L− cells (**d,f**) within the CD45.2+ (**c,d**) and CD45.1+ (**e, f**) subsets in the spleens of recipient *Rag2−/−* mice injected with a mix of either WT:WT or WT:NCOR1-cKO BM cells. (**b**) Numbers indicate the percentages of cells in the respective gates or quadrants. (**c–f**) Cells were pre-gated on CD4+ T cells. Each symbol indicates one mouse. Horizontal bars indicate the mean. *p<0.05, **p<0.01, and ***p<0.001; unpaired two-tailed Student's t-test. Data are representative (**b**) or show a summary (**c–f**) of ten mice that were analyzed in two independent experiments.

The online version of this article includes the following source data and figure supplement(s) for figure 3:

**Figure supplement 1.** Treg cell-specific deletion of NCOR1 results in enhanced CD44hiCD62L− effector Treg cell subsets.

**Figure supplement 1—source data 1.** Raw, uncropped agarose gel picture showing *Ncor1* deletion PCR indicating *Ncor1* deletion.

**Figure supplement 1—source data 2.** Raw, uncropped agarose gel picture showing *Ncor1* deletion PCR indicating *Ncor1* deletion.

cells cultured in Treg cell-inducing conditions with a CD45RA+CD45RO− naïve phenotype (*Figure 2e and f*), although no change was detected in the frequency of CD45RA−CD45RO+ effector Treg cells after NCOR1 deletion. Moreover, CD27, a marker associated with a naïve phenotype in CD4+ T cells (*De Jong et al., 1992*), was downregulated in NCOR1 knockout human CD4+ T cells (*Figure 2e and f*). Together, these data indicate a cross-species conservation of NCOR1 function in regulating naïve and effector Treg cell subset differentiation in mouse and human CD4+ T cells. Furthermore, the data indicate that NCOR1 deletion in human CD4+ T cells enhances the in vitro generation of FOXP3+ T cells.

## NCOR1 regulates naïve and effector Treg cell states in a Treg cell-intrinsic manner

To study whether the changes in naïve and effector Treg cell subset composition is due to a Treg cell-intrinsic requirement for NCOR1, we generated mixed bone marrow (BM) chimeric mice. For this, we transferred a 1:1 mix of either WT (CD45.1+) and *Ncor1*f/f (WT; CD45.2+) or WT (CD45.1+) and NCOR1-cKO (NCOR1-cKO; CD45.2+) BM cells into lethally irradiated *Rag2*−/− mice and analyzed recipient mice 6 weeks after reconstitution (*Figure 3a*). In the spleen of mixed BM chimeric mice, we detected reduced percentages of NCOR1-cKO FOXP3+ Treg cells compared to WT FOXP3+ Treg cells within the CD4+CD45.2+ cell population (*Figure 3b and c*). Within the CD45.2+ FOXP3+ Treg cell population, the CD44hiCD62L− effector Treg cell subset was increased in the absence of NCOR1 compared to the WT CD45.2+ Treg cell population (*Figure 3b and d*). In contrast, there were similar percentages of FOXP3+ and FOXP3+CD44hiCD62L− cells within the CD45.1+ CD4+ T cell population in WT:WT and WT:NCOR1-cKO BM chimeric mice (*Figure 3e and f*). To further exclude the possibility of Treg cell-extrinsic effects on the ratio of naive to effector Treg cells upon *Ncor1* deletion in *Cd4*-Cre mice, we took advantage of the Treg cell-specific *Foxp3*-YFP-Cre deleter strain (*Rubtsov et al., 2008*) to delete *Ncor1* (*Ncor1*f/f *Foxp3*-YFP-Cre mice, designated as NCOR1-cKOFoxp3). Indeed, we observed Treg cell-specific deletion of the *Ncor1* alleles in *Ncor1*f/f YFP+ (i.e. FOXP3+) cells isolated from NCOR1-cKOFoxp3 mice (*Figure 3—figure supplement 1a*), although we cannot formally exclude a stochastic activation of *Foxp3*-Cre in non-Treg cells in some mice as previously reported for other targeted gene loci (*Franckaert et al., 2015*). In line with the results from NCOR1-cKO mice, Treg cell-specific deletion of *Ncor1* resulted in a relative increase in splenic CD44hiCD62L− effector Treg cells compared to WTFoxp3 (*Ncor1*+/+*Foxp3*-YFP-Cre) mice (*Figure 3—figure supplement 1b and c*), while the total cell numbers of CD44hiCD62L− effector NCOR1-cKOFoxp3 Treg cells were slightly but not significantly increased (*Figure 3—figure supplement 1d*). In contrast to *Ncor1* deletion in *Cd4*-Cre mice, deletion of *Ncor1* in *Foxp3*-Cre mice did not alter the frequencies or total numbers of CD4+FOXP3+ cells (*Figure 3—figure supplement 1e and f*). Together, these data indicate that the increase of effector Treg cell subsets in NCOR1-cKO mice is due to Treg cell-intrinsic alterations. Moreover, these data also show that NCOR1 controls naïve and effector Treg cell subset composition once cells started to express the lineage-defining transcription factor FOXP3.

## NCOR1 is essential for Treg cell-mediated protection against intestinal inflammation

Next, we analyzed the impact of NCOR1 deletion on Treg cell effector function. We first investigated whether NCOR1 controls the expression of the anti-inflammatory cytokine TGFβ. Ex vivo PMA/ionomycin stimulation revealed an increase in TGFβ expression in NCOR1-cKO Treg cells compared to WT Treg cells (*Figure 4—figure supplement 1a and b*). To assess whether NCOR1-cKO Treg cells display an altered suppressive activity, we performed in vitro suppression assays in which Treg cells are tested for their ability to restrain the proliferation of WT CD4+ T cells activated by dendritic cells (DC, *Figure 4—figure supplement 1c and d*). To facilitate Treg cell isolation, we crossed WT and NCOR1-cKO mice to the DEREG *Foxp3*-eGFP reporter strain, in which GFP expression is driven by *Foxp3* regulatory elements (*Lahl et al., 2007*). There was no difference in the suppressive activity of splenic NCOR1-cKO.DEREG FOXP3+ Treg cells compared to WT.DEREG FOXP3+ Treg cells.

To assess whether the proportional increase in effector Treg cells correlated with an enhanced suppressive function in vivo, we employed a naïve CD4+ T cell adoptive transfer colitis model, in which CD4+ T cell-mediated disease in recipient *Rag2*-deficient mice is accompanied by several pathological changes including body weight reduction, infiltration of immune cells and loss of colonic crypt structure (*Kiesler et al., 2015*). Co-transfer of Treg cells suppresses T cell-mediated autoimmune disease

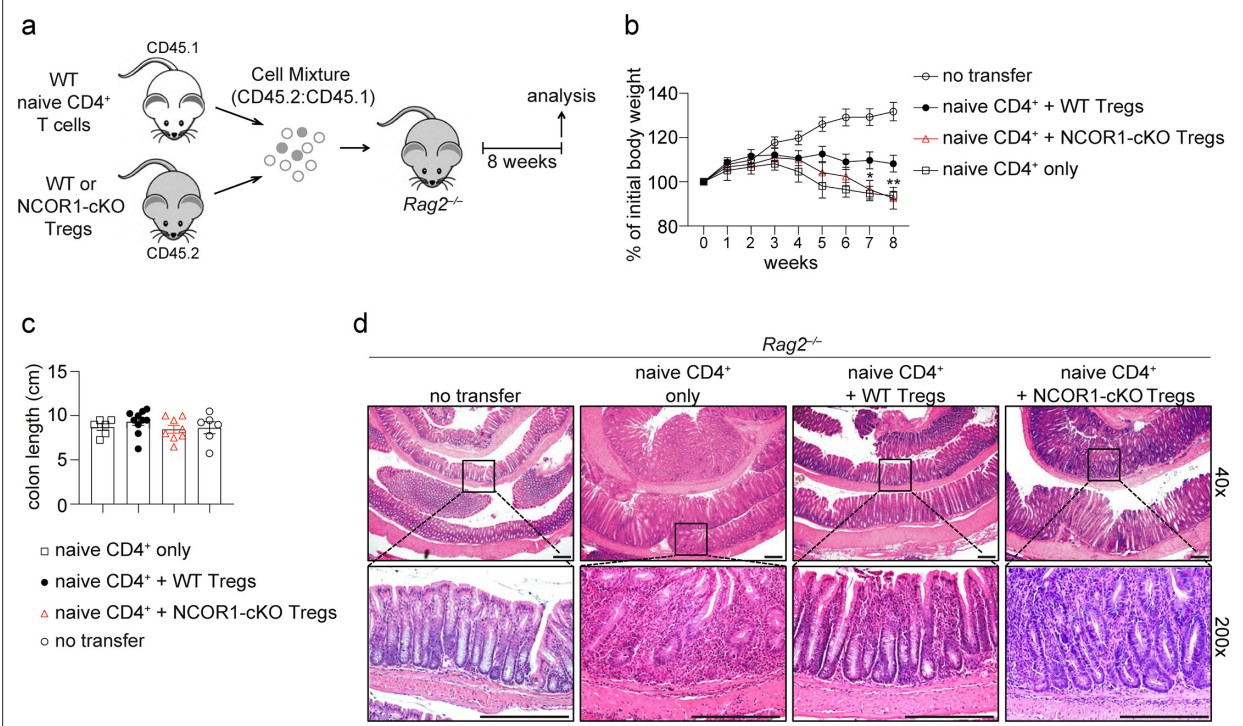

**Figure 4.** NCOR1 is essential for Treg cell-mediated protection in adoptive CD4+ T cell transfer colitis. (**a**) Experimental protocol for adoptive transfer colitis. Control groups received naïve CD4+ T cells only or no cells at all (not shown). (**b**) Weight scores in percentages of initial body weight during the course of colitis in *Rag2−/−* recipient mice are shown. Data show the summary of at least eight mice (except control groups with six mice) of three independent experiments. *p<0.05, **p<0.01, and ***p<0.001 (two-way ANOVA analysis followed by Tukey's multiple-comparisons test). For simplicity, significant differences are shown only between mice that received wild-type (WT) and NCOR1-cKO Treg cells. Of note, non-injected control mice gained significantly more weight (starting from around week 5) compared to all other groups. (**c**) Summary showing colon length from the various *Rag2−/−* recipient mice. (**d**) Colon swiss rolls were processed for hematoxylin and eosin (H&E) staining. The pictures in the bottom represent a 5x magnification of the black rectangle framed section in the top pictures. Magnification: 40x and 200x. Scale bar = 100μm. One representative picture is shown per condition.

The online version of this article includes the following figure supplement(s) for figure 4:

**Figure supplement 1.** In vitro characterization of wild-type (WT) and NCOR1-cKO Treg cells.

and prevents immune pathology (*Powrie et al., 1993*). Therefore, naïve WT CD4+FOXP3− T cells (CD45.1+) were transferred into *Rag2−/−* mice together with Treg cells that were isolated from either WT.DEREG or NCOR1-cKO.DEREG mice (CD45.2+) and recipient mice were monitored over a period of 8 weeks (*Figure 4a*). Mice that co-received NCOR1-cKO Treg cells lost significantly more weight compared to mice co-receiving WT Treg cells, and the weight loss upon transfer of NCOR1-deficient Treg cells was similar to mice that received naïve CD4+ T cells only (*Figure 4b*). Although colon length was not significantly different in any of the groups analyzed (*Figure 4c*), the increased weight loss of mice co-transferred with NCOR1-cKO Treg cells also correlated with a more severe colonic inflammation compared to mice co-transferred with WT Treg cells, indicated by a disruption of crypt structures and enhanced T cell infiltration (*Figure 4d*). Together, these data indicate that NCOR1 is required for Treg cell function in vivo.

To understand the cellular basis of why NCOR1-deficient Treg cells failed to protect against colitis, we characterized lymphocyte subsets in recipient *Rag2−/−* mice including small intestinal intraepithelial lymphocytes (SI-IEL), lamina propria cells (SI-LP) as well as lymphocytes from spleen and mLNs. The frequencies of NCOR1-deficient FOXP3+ Treg cells trended towards a reduction within the SI-IEL and SI-LP cell populations, although total Treg numbers were comparable in mice receiving WT or NCOR1-cKO Treg cells (*Figure 5a and b*). NCOR1-cKO Treg cell (CD45.2+) frequencies were significantly reduced in spleen and mLNs (*Figure 5b*). This was accompanied by slightly increased frequencies of effector CD4+ T cells (CD45.1+) in the spleen and mLNs of mice that received NCOR1-deficient Treg cells compared to mice that received WT Treg cells (CD45.2+) (*Figure 5c*). However, we detected

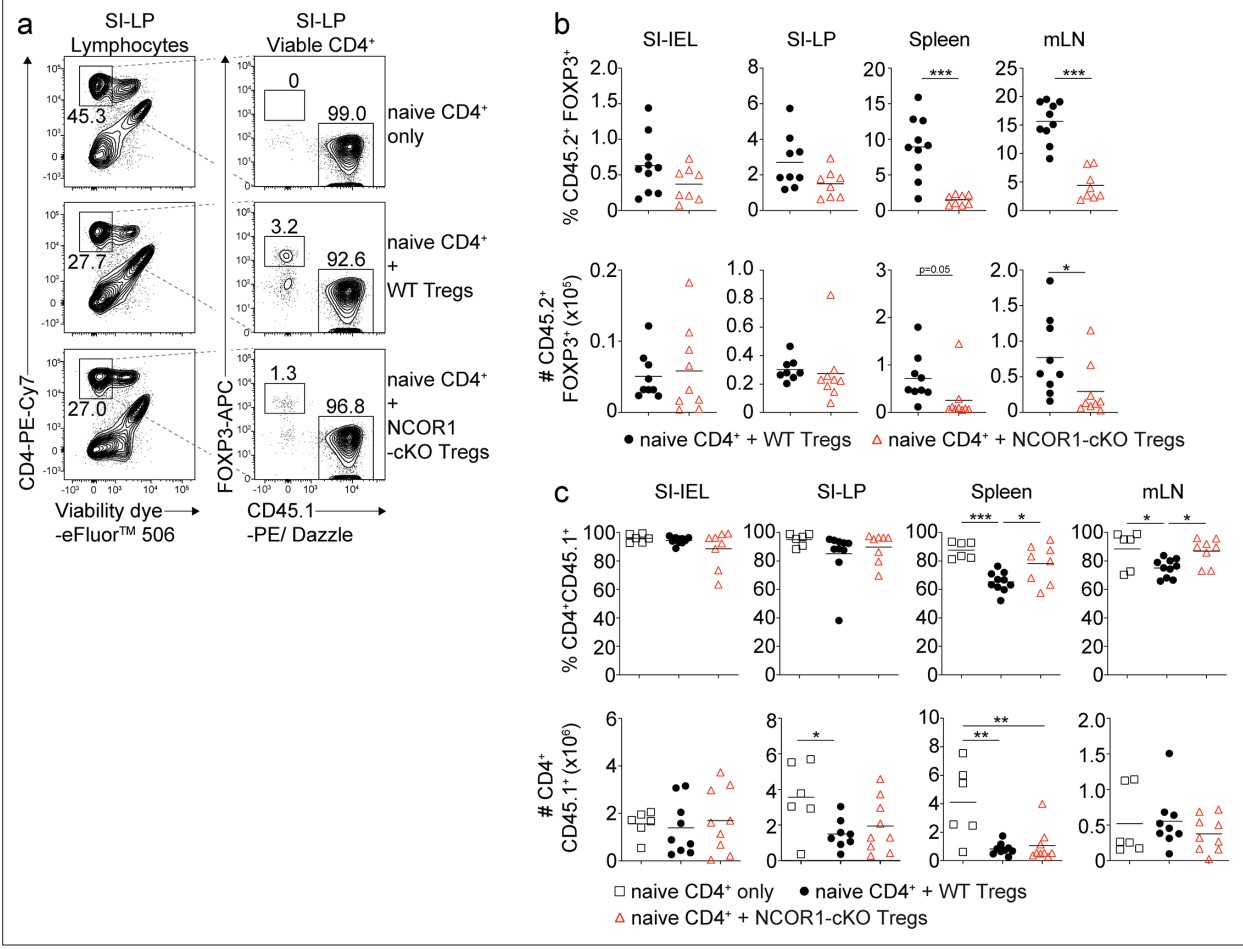

**Figure 5.** Similar frequencies and numbers of wild-type (WT) and NCOR1-cKO Treg cells within small intestinal intraepithelial lymphocyte (SI-IEL) and lamina propria cell (SI-LP) populations in adoptive CD4+ T cell transfer colitis. (**a**) Flow cytometric analysis of SI-LP cells isolated from recipient *Rag2*−/− mice injected with WT CD45.1+CD4+ T cells only (top panel) or with WT CD45.1+CD4+ T cells together with either WT (middle panel) or NCOR1-cKO Treg cells (lower panel). Plots shows the percentage of viable CD4+, CD45.1+FOXP3− and CD45.1−FOXP3+ cells. (**b**) Percentages (upper panel) and numbers (lower panel) of CD45.2+FOXP3+ cells from SI-IEL, SI-LP, spleen, and mLNs of recipient *Rag2*−/− mice injected with WT CD45.1+CD4+ T cells together with either CD45.2+ WT or NCOR1-cKO Treg cells. (**c**) Percentages (upper panel) and numbers (lower panel) of CD4+CD45.1+ cells from SI-IEL, SI-LP, spleen, and mLNs of *Rag2*−/− mice injected with either WT Treg cells, NCOR1-cKO Treg cells, or injected with naïve CD4+ T cells only. (**a**) Numbers indicate the percentages of cells in the respective gate. (**b,c**) Each symbol indicates one mouse. Horizontal bars indicate the mean. *p<0.05, **p<0.01, and ***p<0.001. (**b**) Unpaired two-tailed Student's t-test or (**c**) one-way ANOVA analysis followed by Tukey's multiple-comparisons test. Data are representative (**a**) or show a summary (**b,c**) of at least eight mice (except control groups with six mice) that were analyzed in three independent experiments.

The online version of this article includes the following figure supplement(s) for figure 5:

**Figure supplement 1.** IFNγ and IL17A expression in CD4+ T cells co-transferred with either wild-type (WT) of NCOR1-cKO Treg cells.

no differences in the numbers of effector CD4+ T cells (CD45.1+) (*Figure 5c*) or in the percentages of IL-17A+, IFNγ+, and IL-17A+IFNγ+ expressing CD4+ T cells (CD45.1+) in recipient mice that either received WT or NCOR1-deficient Treg cells (*Figure 5—figure supplement 1a and b*). Although we cannot formally rule out the possibility that lower numbers of NCOR1-cKO Treg cells in the spleen and mLNs contribute to disease development, these results suggest that the failure to protect against colitis might arise from a local functional impairment of NCOR1-deficient Treg cells at earlier time points following T cell transfer.

## NCOR1 ensures transcriptional integrity of naïve and effector Treg cells

As demonstrated above, NCOR1 deletion led to Treg cell-intrinsic alterations that changed the relative abundance of naïve to effector Treg cells. To reveal the underlying NCOR1-dependent

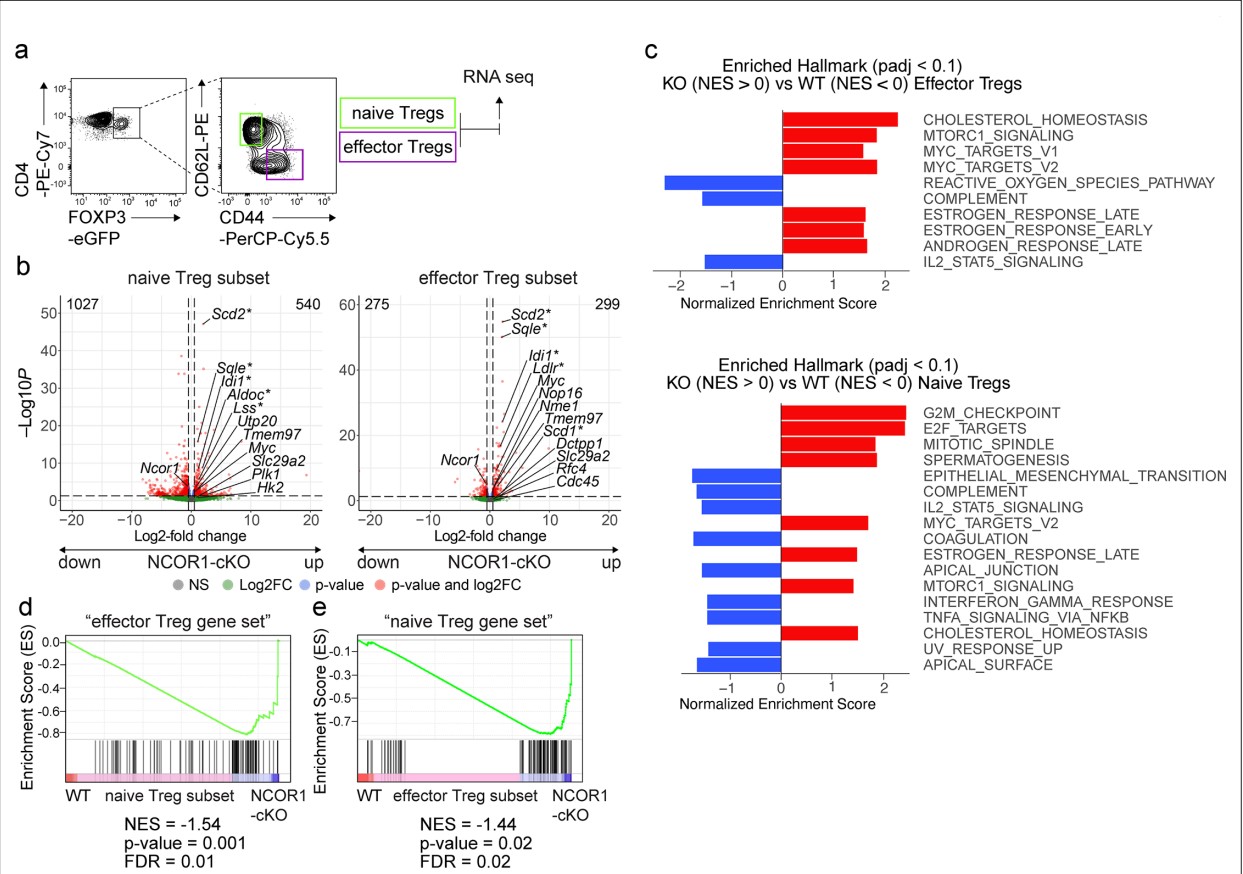

**Figure 6.** NCOR1 controls transcriptional states in naïve and effector Treg cells. (**a**) Contour plots show the gating strategy for the isolation of Treg cells for RNA-sequencing. Cells from spleen and lymph nodes of WT.DEREG and NCOR1-cKO.DEREG mice were isolated and EGFP⁺ (i.e. FOXP3⁺) CD44$^{hi}$CD62L⁻ and CD44$^{lo}$CD62L⁺ cells were sorted and sequenced using the Illumina HiSeq 3000 platform. (**b**) Volcano plots depict a comparison of global gene expression profiles between naïve (CD44$^{lo}$CD62L⁺) WT and NCOR1-cKO Treg cells (left plot) and effector (CD44$^{hi}$CD62L⁻) WT and NCOR1-cKO Treg cells (right plot). On the x-axis log2-fold change is plotted, while the y-axis indicates adjusted p-values (-log10). 1027 genes were downregulated and 540 genes were upregulated in naïve NCOR1-cKO Treg cells. 275 and 299 genes were down- and upregulated, respectively, in effector NCOR1-cKO Treg cells. In addition to *Ncor1* and *Myc*, the top five genes leading to enrichment of cholesterol homeostasis (*) and of Myc targets v2 hallmark gene signatures are shown. (**c**) Diagram showing the top hits of enriched hallmark gene signatures in WT (NES<0) and NCOR1-cKO (NES>0) naïve (top) and effector (bottom) Treg cells. The x axis indicates the Z score. (**d**) Gene set enrichment analysis (GSEA) plots of an 'effector Treg gene set' (containing a list of 100 genes) in naïve NCOR1-cKO Treg cells compared to naïve WT Treg cells. (**e**) Gene set enrichment analysis (GSEA) plots of a 'naïve Treg gene set' (containing a list of 100 genes) in effector NCOR1-cKO Treg cells compared to effector WT Treg cells. (**d,e**) Barcodes indicate the location of the members of the gene set in the ranked list of all genes. NES, normalized enrichment score. The lists of the 'naïve Treg gene set' and the 'effector Treg gene set' are provided in **Supplementary file 1g**.

The online version of this article includes the following figure supplement(s) for figure 6:

**Figure supplement 1.** Deletion of NCOR1 leads to an upregulation of cholesterol pathways.

transcriptional changes, we performed low-input RNA sequencing (RNA-seq) (**Picelli et al., 2014**). To exclude differences in transcriptomes due to the different ratio of naïve to effector populations in WT and NCOR1-cKO Treg cells, we sorted the same number of naïve (CD44$^{lo}$CD62L⁺) and effector (CD44$^{hi}$CD62L⁻) Treg cell subsets from the corresponding DEREG mice and sequenced them separately (**Figure 6a**). We detected a total of 1567 genes being differentially expressed (FDR ≤0.05) in NCOR1-deficient naive Treg cells compared to WT naïve Treg cells, with 540 genes up- and 1027 genes downregulated in the absence of NCOR1 (**Figure 6b** and **Supplementary file 1a**). RNA-seq of effector Treg cell subsets revealed that 574 genes were differentially expressed (FDR ≤0.05) between NCOR1-deficient and WT effector Treg cells. Of these genes, 299 were upregulated and 275 were downregulated in the absence of NCOR1 (**Figure 6b** and **Supplementary file 1b**). To identify transcriptional changes related to well-defined biological states or processes that are altered in the absence of

NCOR1, a Gene Set Enrichment Analysis (GSEA) (*Subramanian et al., 2005*) using the hallmark gene sets of the Molecular Signatures Database (MSigDB) was performed. This uncovered several gene sets that were either enriched or underrepresented in the absence of NCOR1 (a list of all identified hallmark gene sets from this analysis for naïve and effector cells can be found in *Supplementary file 1c* and *Supplementary file 1d*, respectively). Among the altered signatures we identified an enrichment of 'MYC target genes v.2' gene sets, MTORC1 signaling,' 'Estrogen response,' 'E2F targets,' and 'cholesterol homeostasis' in NCOR1-cKO naïve Treg cells compared to their WT counterparts (*Figure 6c*) and these gene sets were also enriched in effector NCOR1-cKO Treg cells (*Figure 6c*). Among the hallmark gene sets that were underrepresented in the absence of NCOR1 was 'IL2_Stat5_signaling' (*Figure 6c*). To further assess transcriptional changes in the absence of NCOR1, a pathway analysis was performed (*Krämer et al., 2014*). This also indicated a strong upregulation of pathways and genes associated with cholesterol biosynthesis in both naïve and effector NCOR1-cKO Treg cell subsets (*Figure 6—figure supplement 1a and b*, *Supplementary file 1e* and *Supplementary file 1f*), similar to observations previously made in conventional naïve NCOR1-cKO CD4+ T cells (*Hainberger et al., 2020*). Of note, the upregulation of the 'Cholesterol homeostasis' hallmark gene set in NCOR1-deficient naive Treg cells (*Figure 6c*) did not lead to accumulation of membrane cholesterol as revealed by comparable Filipin III staining intensities between NCOR1-cKO and WT CD4+CD25+ Treg cells (*Figure 6—figure supplement 1c*). However, the ATP-binding cassette transporters *Abca1* and *Abcg1*, important regulators of cholesterol efflux (*Tarling and Edwards, 2012*), were upregulated in naïve (*Abca1*) and effector (*Abca1, Abcg1*) NCOR1-cKO Treg cells (*Figure 6—figure supplement 1d*). Thus, we conclude that despite an elevated cholesterol biosynthesis pathway in NCOR1-cKO Treg cells, there is no accumulation of membrane-inserted cholesterol, likely due to enhanced cholesterol export caused by increased expression of *Abca1* and *Abcg1*.

Since the proportion of effector Treg cells is increased in NCOR1-cKO mice, we next addressed the question of whether NCOR1 deficiency in naïve Treg cells leads to the induction of transcriptional signatures characteristic of effector Treg cell subsets. Therefore, we defined an 'effector Treg gene set' based on our RNA-seq data that contained the top 100 genes upregulated (based on FC differences, FDR ≤0.05) in effector WT Treg cells compared to naïve WT Treg cells (see *Supplementary file 1g* for the list of genes). GSEA with this pre-defined gene list revealed enrichment of the 'effector Treg gene set' in naïve NCOR1-cKO Treg cells compared to naïve WT Treg cells (*Figure 6d*). This indicates that *Ncor1* deletion leads to an upregulation of an effector gene signature in naïve NCOR1-cKO Treg cells. Moreover, GSEA also revealed an enrichment of a 'naïve Treg gene set' (i.e. the top 100 genes upregulated in WT naïve Treg cells compared to WT effector Treg cells) in effector NCOR1-cKO Treg cells (*Figure 6e*). Taken together, our RNA-seq analysis indicates that naïve CD44loCD62L+ Treg cells lacking NCOR1 inherently display an effector Treg signature and vice-versa. Thus, NCOR1 is essential for establishing and/or maintaining transcriptional programs that define and potentially regulate naïve and effector Treg cell states.

## NCOR1 regulates naïve and effector Treg cell states in a LXRß-independent manner

The nuclear receptor liver X receptor (LXR) acts as a cholesterol sensor and is a crucial regulator of cholesterol biosynthesis (*Zhao and Dahlman-Wright, 2010*). Two isoforms of LXR, namely LXRα (encoded by the *Nr1h3* gene) and LXRβ (encoded by *Nr1h2*) are known (*Willy et al., 1995*; *Shinar et al., 1994*). Naïve and effector WT Treg cells expressed high levels of *Nr1h2* but virtually no *Nr1h3*, and this isoform-specific expression pattern was maintained in NCOR1-cKO Treg cells compared to WT Treg cells (*Figure 7—figure supplement 1a*). In the absence of LXR ligands, NCOR1 interacts with LXR and inhibits the expression of LXR target genes (*Mottis et al., 2013*; *Li et al., 2013*). Upon LXR ligand binding, NCOR1 dissociates from LXR leading to its activation and the expression of LXR target genes (*Hu et al., 2003*). Due to the observed upregulation of hallmark gene sets 'Cholesterol homeostasis' (*Figure 6c*) and LXR/RXR activation pathways (*Figure 6—figure supplement 1a*) in NCOR1-cKO Treg cells, we investigated whether a disrupted LXRβ-NCOR1 interaction in Treg cells results in enhanced effector Treg differentiation using two experimental strategies based on pharmacological and genetic approaches. First, to test whether LXR activation contributed to the relative increase in effector Treg cell subsets, we differentiated naïve WT CD4+ T cells into in vitro induced Treg cells (iTreg cells) in the presence of the LXR agonist GW3965 (*Collins et al., 2002*).

GW3965 treatment did not alter the percentage of FOXP3$^+$ cells, however, resulted in a small but significant increase in the fraction of CD44$^{hi}$ cells (*Figure 7—figure supplement 1b*). GW3965 treatment mildly affected cell viability and proliferation at low concentrations (up to 4 µM), but showed more severe effects at a higher concentration (8 µM) (*Figure 7—figure supplement 1b*). Next, to dissect a potential functional interaction between NCOR1 and LXRβ in the induction of effector Treg cells in vivo, we employed a genetic approach and intercrossed NCOR1-cKO mice with LXRβ-cKO (*Nr1h2*$^{f/f}$) mice (*Michaels et al., 2021*) to generate NCOR1-LXRβ-cDKO double-deficient mice. To obtain a sufficiently high number of mice for the analysis, we established independent breeding colonies for NCOR1-cKO, LXRβ-cKO as well as for NCOR1-LXRβ-cDKO mice and used the corresponding *Cre*-negative littermates as WT controls (i.e. *Ncor1*$^{f/f}$, designated as WT$^{Ncor1}$; *Nr1h2*$^{f/f}$, as WT$^{Lxrb}$; and *Ncor1*$^{f/f}$,*Nr1h2*$^{f/f}$ as WT$^{Ncor1/Lxrb}$, respectively). In agreement with a previous study (*Michaels et al., 2021*), we observed a strong reduction in the frequencies of splenic T cells in the absence of LXRβ (*Figure 7—figure supplement 2a and b*). Within the TCRβ$^+$ population, CD4$^+$ T cells were slightly reduced, while the frequency of CD8$^+$ T cells was not altered (*Figure 7—figure supplement 2b and c*). Of note, NCOR1-LXRβ-cDKO mice also exhibited a severe reduction of T cell numbers and slightly lower frequencies of CD4$^+$ T cells as observed in LXRβ-cKO mice (*Figure 7—figure supplement 2a, b and c*). Interestingly, the Treg cell frequency within the LXRβ-cKO CD4$^+$ T cell population in the spleen was higher compared to WT littermate controls (*Figure 7a and b*), in contrast to NCOR1-cKO mice where a twofold relative decrease was observed (*Figure 7a and b*). NCOR1-LXRβ-cDKO mice also exhibited a higher representation of Treg cells amongst the CD4$^+$ T cell population (*Figure 7a and b*), suggesting that deletion of LXRβ is sufficient to rescue the low Treg cell frequency phenotype seen in NCOR1-cKO mice. Within the LXRβ-cKO Treg cell population, there was a tendency of a mild increase in CD44$^{hi}$CD62L$^-$ effector cell frequencies (p=0.058), while CD44$^{lo}$CD62L$^+$ naïve Treg cells were not changed (*Figure 7c and d*). This suggests that LXRβ deletion (using CD4-*Cre)* had only a small effect on the ratio of naïve to effector Treg cells. In contrast, splenocytes from NCOR1-LXRβ-cDKO mice displayed a strong increase in the frequency of CD44$^{hi}$CD62L$^-$ effector Treg cells along with a severe decrease in naïve Treg cells (*Figure 7c and d*), similar to NCOR1-cKO mice. This indicates that loss of LXRβ did not revert alterations in the ratio of naïve to effector Treg cells induced by NCOR1 deletion. On the contrary, within the Treg cell population of some splenocytes from NCOR1-LXRβ-cDKO mice more than 50% of effector Treg cells were detected (*Figure 7c and d*), a frequency not reached in NCOR1-cKO mice (*Figures 1b and 7d*). This might indicate a minor contribution of LXRβ in the generation of effector Treg cells in the absence of NCOR1.

The transcription factor MYC plays a key role in the transition to an effector Treg cell state, as well as in the metabolic programming of effector Treg cells (*Saravia et al., 2020*; *Angelin et al., 2017*). NCOR1 also has been shown to cooperate with MYC (*Zhuang et al., 2018*). We observed an induction of Myc target genes (V1 and V2 gene sets) in the absence of NCOR1 (*Figure 6c*) and our RNA-seq data revealed that *Myc* expression was upregulated both in naïve and in effector NCOR1-cKO Treg cells in comparison to the corresponding WT subsets (*Figure 6b*, *Supplementary file 1c* and *Supplementary file 1d*). Therefore, in the next step, we investigated whether MYC protein expression levels were affected by the absence of NCOR1, LXRβ, or both. Intracellular staining revealed no difference in MYC protein levels between WT$^{Ncor1}$ and NCOR1-cKO naïve Treg cells and slightly lower MYC levels in effector Treg cells (*Figure 7—figure supplement 3a*), contrasting with the observed upregulation of *Myc* mRNA levels and MYC target genes. MYC protein expression was significantly increased in naïve CD44$^{lo}$CD62L$^+$ LXRβ-cKO and NCOR1-LXRβ-cDKO FOXP3$^+$ Treg cells in comparison to the respective WT$^{Lxrb}$ and WT$^{Ncor1/Lxrb}$ controls (*Figure 7—figure supplement 3a*), suggesting that loss of LXRβ has a dominant effect on MYC levels in this Treg cell subset. However, elevated MYC protein levels in LXRß-deficient Treg cells were not sufficient to induce an increase in effector Treg cells. This suggests that NCOR1 might restrain MYC activity, which in turn might contribute to the observed increase in effector Treg cells upon NCOR1 deletion. Of note, while LXRβ-cKO mice also displayed higher MYC protein levels in effector Treg cells relative to their WT counterparts, this was reverted in NCOR1-LXRβ-cDKO animals (*Figure 7—figure supplement 3a*). We also assessed how CRISPR-Cas9 mediated-knockout of NCOR1 affected MYC expression levels in human CD4$^+$ T cells cultured in Treg cell-inducing conditions (*Figure 2d*). Deletion of NCOR1 led to an increase in the fraction of human CD4$^+$ T cells expressing high levels of MYC (*Figure 7—figure supplement 3b and c*). The MYC$^{hi}$ population of human CD4$^+$ T cells contained

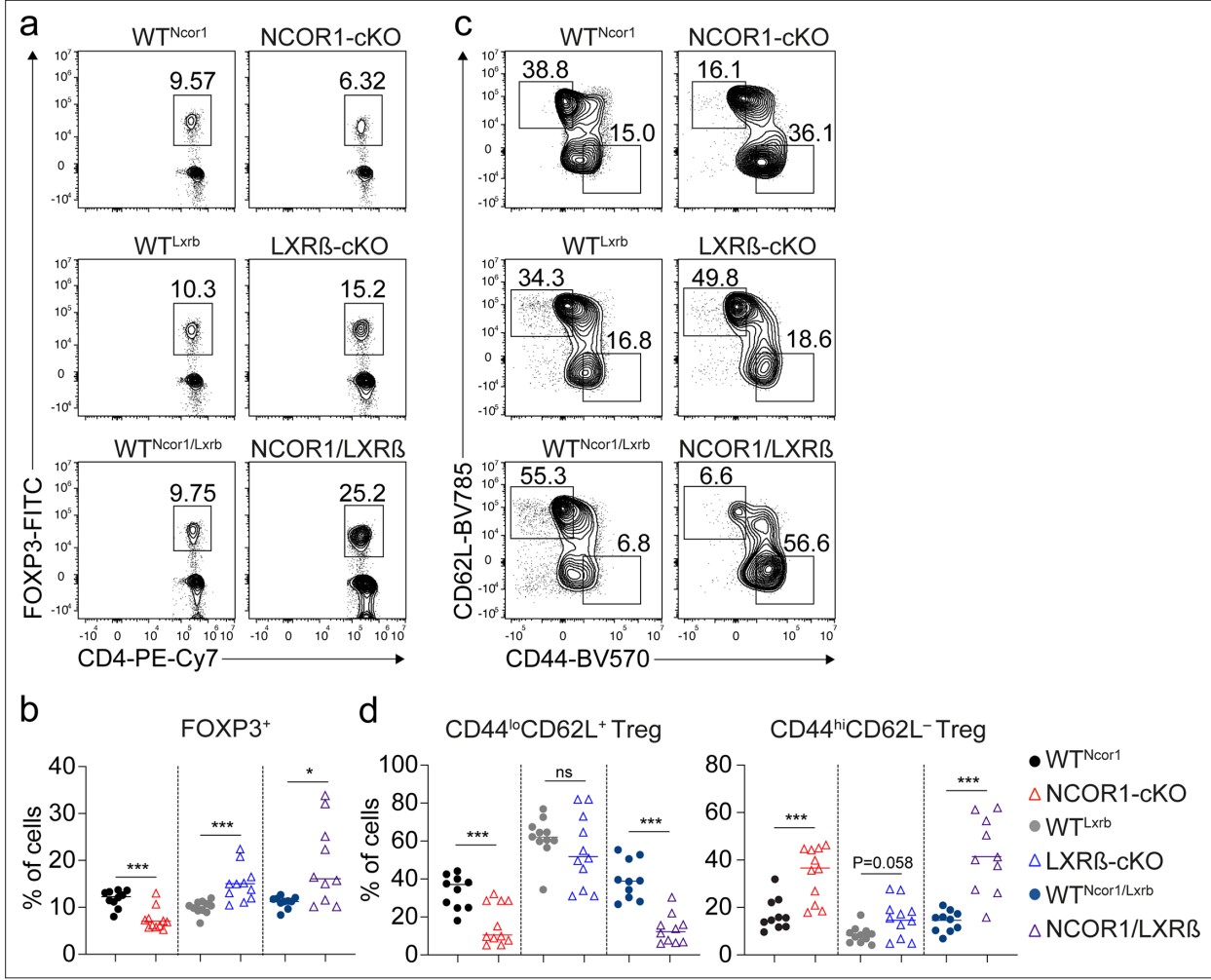

**Figure 7.** NCOR1 controls the ratio of naïve to effector Treg cells in an LXRß-independent manner. (**a**) Contour plots show FOXP3 and CD4 expression in splenic TCRβ⁺CD4⁺ cells of mice of the indicated genotype. (**b**) Percentages of FOXP3⁺ CD4⁺ T cells of all mice analyzed as described in (**a**). (**c**) Flow cytometric analysis of splenic FOXP3⁺ CD4⁺ T cells from mice of the indicated genotype showing CD44 and CD62L expression at steady-state. (**d**) Percentage of CD44^lo^CD62L⁺ (left) and CD44^hi^CD62L⁻ (right) Treg cells of all mice analyzed as described in (**c**). (**a,c**) Numbers next to the regions indicate cell frequencies. (**b,d**) Each symbol indicates one mouse. Horizontal bars indicate the mean. *p<0.05, **p<0.01, and ***p<0.001. (**b,d**) Unpaired two-tailed Student's t-test comparing the respective WT and knockout groups. Data are representative (**a,c**) or show the summary (**b,d**) of at least ten mice that were analyzed in at least eight independent experiments.

The online version of this article includes the following figure supplement(s) for figure 7:

**Figure supplement 1.** LXRβ agonist-treatment of WT iTreg cells leads to a small increase in CD44^hi^ Treg cells.

**Figure supplement 2.** Phenotypic characterization of NCOR1/LXRβ-cDKO mice.

**Figure supplement 3.** MYC protein expression levels in NCOR1-cKO, LXRβ-cKO and NCOR1-LXRβ-cDKO Treg cell subsets and in human NCOR1-knockout CD4⁺ T cells cultured under iTreg cell differentiation conditions.

**Figure supplement 4.** NCOR1 positively regulates FOXP3⁺ Treg cell differentiation and controls naïve and effector Treg cell subset integrity.

a higher percentage of FOXP3-expressing cells in the absence of NCOR1 when compared to CD4⁺ T cells treated with non-targeting (NT)-sgRNAs (*Figure 7—figure supplement 3b and c*). Similar to murine Treg cells, MYC protein levels were similar between NCOR1-deficient and WT human FOXP3⁺ CD4⁺ T cells.

Taken together, these data indicate that NCOR1 controls naïve and effector Treg cell states, and the relative distribution of Treg cell subsets in an LXRβ-independent manner. Moreover, our data also suggest that NCOR1 restrains MYC activity, which might contribute to the observed increase in effector Treg cells upon NCOR1 deletion.

## Discussion

In this study, we demonstrate that the transcriptional regulator NCOR1 is a key factor controlling the ratio of naïve to effector Treg cells at steady state and during immune responses in a Treg cell-intrinsic manner. We, and others, have previously reported that loss of NCOR1 in T cells using *Cd4*-Cre results in developmental alterations during positive/negative selection, as well as in reduced survival of single-positive cells (*Wang et al., 2017*; *Müller et al., 2017*). The Treg-specific deletion of *Ncor1* using the *Foxp3*-Cre deleter strain resulted in an increase in effector Treg cells as observed in mice with a pan T cell-specific deletion using *Cd4*-Cre. However, in contrast to *Ncor1*[f/f] *Cd4*-Cre mice (*Müller et al., 2017*), there was no overall decrease in the percentage of the FOXP3[+] T cell population in NCOR1-cKO[Foxp3] mice compared to WT[Foxp3] mice at steady state. This suggests that Treg cell survival is not dependent on NCOR1 once Treg cell lineage differentiation has been initiated. These data, therefore, rule out the possibility that changes in the frequencies of naïve and effector subsets within the Treg cell population are due to positive/negative selection defects or due to the reduced survival of single-positive cells in NCOR1-cKO mice (*Wang et al., 2017*; *Müller et al., 2017*). Therefore, our data indicate that NCOR1 controls the establishment of Treg cell subset homeostasis once *Foxp3* expression, and thus Treg cell lineage differentiation, has been induced.

Effector Treg cell subsets have a higher suppressive activity and are crucial for maintaining immune homeostasis in peripheral tissues (*Liston and Gray, 2014*). Therefore, one might have expected that the increase in effector Treg cells results in an enhanced suppressive function of the NCOR1-deficient Treg cell population. Interestingly, there was no difference in the suppressive activity between WT and NCOR1-cKO Treg cells in vitro. In addition, in an adoptive CD4[+] T cell transfer colitis model, co-transferred NCOR1-cKO Treg cells even failed to protect recipient mice from weight loss and intestinal tissue damage in comparison to co-transferred WT Treg cells. We observed an overall decrease of NCOR1-deficient Treg cells in recipient mice, although the extent of the reduction varied among various lymphoid tissues. This might indicate either that NCOR1 is part of a regulatory network that integrates lymphoid tissue-specific signals for tissue-specific Treg cell maintenance, or that NCOR1 controls the ability of Treg cells to home to different tissues, at least in an adoptive transfer setting. We detected a strong decrease in NCOR1-cKO Treg cells in mLNs after transfer. Since it has been shown that Treg cells home to mLNs early after transfer and proliferate there until colon inflammation is resolved (*Mottet et al., 2003*), one might speculate that the failure of NCOR1-cKO Treg cells to prevent colitis is caused by the reduction in Treg numbers in gut-draining LNs. However, intestinal NCOR1-cKO Treg cell numbers were not altered within the SI-IEL population and SI-LP cells, suggesting that the disease was aggravated (at least in part) due to an impaired suppressive function of Treg cells in the absence of NCOR1. Further evidence that effector Treg cells might not be fully functional comes from our RNA-Seq analysis. Our GSEA data indicated an upregulation of an 'effector Treg gene set' in naïve Treg cells, thus potentially a 'priming' of naïve NCOR1-deficient Treg cells towards the acquisition of an effector phenotype. However, a 'naïve Treg gene set' was enriched in NCOR1-deficient effector Treg cells. Thus, NCOR1-cKO Treg cells that upregulated CD44 (CD44[hi]) and downregulated CD62L (CD62L[-]), and hence are classified as effector Treg cells, still have transcriptional features of naïve Treg cells in comparison to WT effector Treg cells. Consequently, NCOR1-cKO CD44[hi]CD62L[-] Treg cells might not represent fully functional effector Treg cells. This suggests that NCOR1 is essential in establishing naïve and effector Treg cell states.

Another important finding of our study is that NCOR1 controls naïve and effector Treg states in a LXRβ-independent manner. LXRβ is a key transcription factor of the nuclear receptor family regulating the expression of genes required for cholesterol and lipid metabolism. In the absence of activating ligands, LXRβ target genes are repressed due to the interaction of LXRβ with NCOR1 and NCOR1-associated repressor complexes that contain chromatin-modifying molecules such as HDAC3 (*Zhao and Dahlman-Wright, 2010*; *Mottis et al., 2013*; *Li et al., 2013*). LXRβ regulates T cell proliferation and function and is crucial for maintaining cholesterol homeostasis in T cells (*Bensinger et al., 2008*). The observed upregulation of several pathways associated with cholesterol biosynthesis in NCOR1-deficient Treg cells pointed to an activation of LXRβ-mediated transcriptional circuits and hence to the induction of (some) LXRβ target genes. This might be linked to the enhanced effector Treg differentiation, substantiated by the observation that LXR agonist GW3965 treatment of WT iTreg cells phenocopied the increased expression of CD44 as observed in NCOR1-cKO Treg cells and in human CD4[+] T cells with an NCOR1 knockout. However, our genetic data demonstrated that

deletion of LXRβ in the T cell lineage did not recapitulate the increase in CD44$^{hi}$ CD62L$^{-}$ effector Treg cells in the spleen observed in NCOR1-cKO mice. The combined deletion of NCOR1 and LXRβ resulted in a strong increase in LXRβ-NCOR1 effector Treg cells and a corresponding severe reduction in naïve Treg cells, similar to the phenotype observed in NCOR1-cKO mice. This strongly suggests that NCOR1 controls the relative composition of naïve and effector Treg cell subsets in a LXRß-independent manner. On the other hand, some LXRβ-NCOR1 mice displayed more than 50% of CD44$^{hi}$CD62L$^{-}$ effector cells among the Treg cell population, which was not observed in NCOR1-cKO mice. This suggests that LXRβ might have a minor role in these processes, at least in the absence of NCOR1. Of note, and as previously reported (*Michaels et al., 2021*), LXRβ deficiency resulted in a relative increase in FOXP3$^{+}$ T cells among the CD4$^{+}$ T cell population, while FOXP3$^{+}$ T cells within the CD4$^{+}$ T cell lineage was reduced in the absence of NCOR1. LXRβ deletion on top of NCOR1 deletion reverted the relative decrease in Treg cells, which suggests that NCOR1 controls the generation of Treg cells in an LXRβ-dependent manner. NCOR1 suppresses LXRβ or counteracts LXRβ-induced pathways, which itself restrains the generation of FOXP3$^{+}$ Treg cells. Loss of NCOR1 potentially enhances LXRβ activity or LXRβ-induced pathways, thereby reducing Treg cell frequency. In contrast, deletion of LXRβ (in the presence or absence of NCOR1) abolishes the restraining activity of LXRβ-induced pathways, resulting in increased frequencies of Treg cells among the CD4$^{+}$ T cell population (*Figure 7—figure supplement 4*). Our RNA-seq also revealed an upregulation of MYC target genes in NCOR1-cKO Treg cells which correlated with increased *Myc* gene expression, and we even observed an upregulation of MYC protein expression by flow cytometry in LXRβ and NCOR1/LXRβ-deficient T cells. MYC is an essential driver for the differentiation of naive Treg cells into effector cells (*Saravia et al., 2020*). However, elevated MYC protein levels in LXRß-deficient Treg cells were not sufficient to induce an increase in effector Treg cells. This suggests that NCOR1 might restrain MYC activity and that a dysregulated MYC activity might be one of the contributing factors leading to a relative increase in effector Treg cells in the absence of NCOR1.

It must be acknowledged that our study has some limitations. Further investigations are needed to understand to which extent NCOR1 controls effector Treg differentiation in an LXRβ-independent and LXRβ-dependent manner. It has also been reported that LXRβ-deficient Treg cells, in a bone marrow chimeric setting, exhibit decreased expression of CD44 and increased expression of CD62L (*Michaels et al., 2021*). However, this decrease in effector Treg cells (based on CD44 and CD62L expression) might be an indirect effect, since LXRβ also controls fitness and functionality of activated Treg cells (*Michaels et al., 2021*). Additional studies are warranted to determine how NCOR1 and LXRβ containing complexes control Treg homeostasis, naive to effector Treg cell ratios as well as effector Treg differentiation and function. These might include experiments to determine NCOR1 and LXRβ target binding sites in order to elucidate which target genes are (co-)regulated by NCOR1 and LXRβ during FOPX3$^{+}$ Treg cell differentiation, and whether T cell activation disrupts NCOR1 and LXRβ interaction. Moreover, it has not been addressed whether NCOR1 modulates the activity of other nuclear receptors and transcription factors in Treg cells or how this might contribute to naïve and effector Treg cell states. In addition, NCOR1 controls Th1 and Th17 cells (*Hainberger et al., 2020*), however, it remains to be determined whether NCOR1 is essential for maintaining T cell function in general or whether Th cell subset specific functions exist.

In summary, here we report that NCOR1 is a key member of the transcriptional network that controls FOXP3$^{+}$ regulatory T cell subset differentiation and function. NCOR1 antagonizes LXRß-induced pathways that restrain the generation of FOXP3$^{+}$ Treg cells, while NCOR1 controls the relative abundance of naïve and effector Treg cell subset independently of LXRß. Moreover, NCOR1 is essential for the proper suppressive function of Treg cells. We propose that NCOR1 is essential for maintaining naïve and effector Treg cell states by ensuring the integrity of naïve and effector transcriptional programs in FOXP3$^{+}$ Treg cells, as well as linking their proper differentiation with their functionality.

## Materials and methods

### Animal models

Animal experiments were evaluated by the ethics committees of the Medical University of Vienna and approved by the Austrian Federal Ministry for Education, Science and Research (GZ:BMB-WF-66.009/0039 V/3b/2019, GZ:BMBWF-66.009/0326 V/3b/2019). Animals were maintained in

research facilities of the Department for Biomedical Research at the Medical University of Vienna. Animal husbandry and experiments were performed under national laws in agreement with guidelines of the Federation of European Laboratory Animal Science Associations (FELASA), which correspond to Animal Research: Reporting of In Vivo Experiments from the National Center for the Replacement, Refinement, and Reduction of Animals in Research (ARRIVE) guidelines. *Ncor1*$^{f/f}$ mice (*Yamamoto et al., 2011*) were kindly provided by Johan Auwerx. *Cd4*-Cre mice (*Lee et al., 2001*) were kindly provided by Chris Wilson. *Foxp3*-Cre mice (*Rubtsov et al., 2008*) were kindly provided by Alexander Rudensky. DEREG mice (*Lahl et al., 2007*) were kindly provided by Tim Sparwasser. CD45.1$^+$ and *Rag2*$^{-/-}$ mice were kindly provided by Jochen Hühn. Mice analyzed were 8–12 weeks of age and of mixed sex, except *Foxp3*-Cre mice of which only males were analyzed.

## Genotyping of mice

The PCR for the *Ncor1* deletion (floxed band: 346 bp. delta band: 246 bp) was carried out for 5 min at 96 °C, followed by 39 cycles of 30 s at 94 °C, 30 s at 56 °C, and 1 min at 72 °C. The PCR for the *Cd4*-Cre transgene (300 bp) was carried out for 5 min at 96 °C, followed by 39 cycles of 30 s at 94 °C, 30 s at 56 °C, and 1 min at 72 °C. The PCR for the DEREG (GFP) transgene (450 bp) was carried out for 1 min at 96 °C, followed by 35 cycles of 1 min at 94 °C, 1 min at 60 °C, and 1 min at 72 °C. The PCR for the *Foxp3*-Cre transgene (346 bp) was carried out for 1 min at 94 °C, followed by 34 cycles of 30 s at 94 °C, 30 s at 60 °C, and 45 s at 72 °C. Primer: *Ncor1* floxed and deletion (#31): 5'-TTG GCC TTG GAG TAA ATG CTG TGA G –3'. *Ncor1* floxed and deletion (#32): 5'-GGA AAC TAC CTA CCT GAA TCC ATG G-3'. *Ncor1* deletion (#29): 5'-GAA CTA AGG ACA GGA AGG TAC AGG G-3'. *Cd4*-Cre forward: 5'-TCT CTG TGG CTG GCA GTT TCT CCA-3'. *Cd4*-Cre reverse: 5'-TCA AGG CCA GAC TAG GCT GCC TAT-3'. DEREG forward: GCG AGG GCG ATG CCA CCT ACG GCA-3'. DEREG reverse: 5'-GGG TGT TCT GCT GGT AGT GGT CGG-3'. *Foxp3*-Cre forward: 5'-AGG ATG TGA GGG ACT ACC TCC TGT A-3'. *Foxp3*-Cre reverse: 5'-TCC TTC ACT CTG ATT CTG GCA ATT T-3'. *Nr1h2(Lxrb)* floxed forward: 5'-ACT AAC CCC ACA TTA CCG TGA GGC-3'. *Nr1h2(Lxrb)* floxed reverse: 5'-AGG TGC CAG GGT CTT GC AGT-3'.

## Purification of CD4$^+$ T Cells

Cells from the spleen, axillary, brachial, and inguinal lymph nodes (LN) were isolated, pooled and single-cell suspensions were made using a 70 µm cell strainer (Corning) in a six-well plate (Sarstedt) containing PBS (Sigma) with 2% fetal bovine serum (FCS) (Biowest). Red blood cells were removed using BD Pharm Lyse (BD Biosciences). Cells were resuspended in PBS/FCS containing a master mix of biotinylated antibodies (Gr-1, B220, NK1.1, CD11b, CD11c, CD8α, TER-119, CD44, and CD25). CD4$^+$ T cells were enriched by negative depletion using magnetic streptavidin beads (MagniSort SAV Negative Selection beads; Invitrogen) according to the manufacturer's instructions. Biotinylated antibodies against Ly-6G/Ly-6C (clone: RB6-8C5), CD45R/B220 (clone: RA3-6B2), NK1.1 (clone: PK136), CD11c (clone: N418), CD11b (clone: MEL1/70), CD8α (clone: 53–6.7), TER-119 (clone: TER-119), CD44 (clone: IM7) and CD25 (clone: PC61) were purchased from Biolegend. For some experiments, cells were further sorted into naïve CD4$^+$ T cells (CD25$^-$CD44$^{lo}$CD62L$^+$) on an SH800 cell sorter (SONY).

## Flow cytometric analysis

Isolated T cells and in vitro-cultured T cells were incubated with Fc-block (1:250; BD Biosciences) followed by surface staining. Dead cells were excluded using Fixable Viability Dye eFluor 506 or 780 (Thermo Fisher Scientific) according to the manufacturer's protocol. For intracellular transcription factor and cytokine staining, cells were fixed and permeabilized using the Foxp3 Staining Buffer Set (Thermo Fisher Scientific) according to the manufacturer's protocol and stained with the appropriate antibodies. Cells were additionally stained with CellTrace Violet Proliferation dye (Thermo Fisher Scientific) wherever indicated. Flow cytometry data was acquired using a BD FACS Fortessa flow cytometer (BD Biosciences) and analyzed using FlowJo v10.2 software (TreeStar). Gating strategies are shown in *Figure 1—figure supplement 3*.

## Flow cytometry antibodies

The following anti-mouse antibodies were used for flow cytometry: CD19 (clone: 6D5, Biolegend), CD25 (clone: PC61, BD Biosciences), CD4 (clone: RM4-5, Biolegend), CD44 (clone: IM7, Biolegend),

CD45.1 (clone: A20, Biolegend), CD45.2 (clone: 104, Biolegend), CD62L (clone: MEL-14, Biolegend), CD69 (clone: H1.2F3, Thermo Fisher Scientific), CD8 (clone: 53–6.7, BD Biosciences), CD11c (clone: E418, eBioscience), CTLA4 (clone: UC10-4B9, Biolegend), CXCR5 (clone: L138D7, Biolegend), FOXP3 (clone: FJK-16s, Thermo Fisher Scientific), GITR (clone: REA980, Miltenyi Biotec), ICOS (clone: 7E.17G9, Thermo Fisher Scientific), IFNγ (clone: XMG1.2, Biolegend), IL-17A (clone: TC11-18H10.1, Biolegend), KI67 (clone: 16A8, Biolegend), KLRG1 (clone: 2F1, Biolegend), MYC (clone: Y69, Abcam), PD1 (clone: 29 F.1A12, Biolegend), p-AKT (clone: D9E, Cell Signalling), p-ERK (clone: 20 A, BD Phosflow), p-S6 (clone: D68F8, Cell Signalling), TCRβ (clone: H57-597, Thermo Fisher Scientific), TGFβ (clone: EPR21143, Abcam), CD80 (clone: 16–10 A1, Biolegend), CD86 (clone: GL1, Biolegend), MHCII (clone: M5/114.15.2, eBioscience). The following anti-human antibodies were used for flow cytometry: CD45RA (clone: HI100, Biolegend), CD45RO (clone: UCHL1, Biolegend), CD25 (clone: BC96, Biolegend), CD27 (clone: O323, Biolegend). FOXP3 (clone: PCH101, eBioscience), MYC (clone: Y69, Abcam).

## In vitro suppression assays

The experiment was performed as described previously (*Andersen et al., 2019*). In brief, FACS-purified CD45.1$^+$ naive CD4$^+$CD44$^{lo}$CD62L$^{hi}$ T cells were labeled with 20 µM cell proliferation dye eFluor450 (eBioscience) for 10 min at 37 °C before culture and activated in the presence of 1×10$^5$ antigen-presenting cells (irradiated CD45.2$^+$ splenocytes) and 1 µg/ml anti-CD3ε (BD Biosciences). 5×10$^4$ responder T cells were cultured in U-bottom 96-well plates in T cell medium (supplemented with 1 mM sodium pyruvate and 100 mM non-essential amino acids) at different ratios with FACS-purified CD45.2$^+$ eGFP$^+$ Treg cells isolated from the spleens and LNs of WT.DEREG and NCOR1-cKO. DEREG mice. After 72 hr, cells were harvested and stained with appropriate antibodies and subjected to flow cytometric analysis.

## Adoptive CD4$^+$ T cell transfer colitis

4×10$^5$ flow cytometry-sorted naive (CD25$^-$CD44$^{lo}$CD62L$^+$) CD4$^+$ T cells from CD45.1$^+$ congenic mice were injected *i.p.* into *Rag2$^{-/-}$* mice alone or together with either 1x10$^5$ CD45.2$^+$ flow cytometry-sorted WT.DEREG or NCOR1-cKO.DEREG Treg cells. Control *Rag2$^{-/-}$* mice did not receive any cells. Weight of the mice was monitored over the course of 8 weeks and mice were subsequently sacrificed for organ analysis. T cells were isolated from spleen and mLNs as described above, stained with appropriate antibodies, and subjected to flow cytometric analysis. For SI-LP and SI-IEL isolation, small intestines (SIs) were isolated and stool and mucus were removed. The tissue was transferred into Petri dishes, cut into small pieces, and washed three times by addition of 40 ml wash solution (1 X HBSS, HEPES-bicarbonate (pH 7.2), and 2% FBS) and vortexing for 15 s. The samples were filtered through 100 µm cell strainers and subsequently incubated in 20 ml EDTA solution (10% FBS, 1 X HBSS, 15 mM HEPES, 1 mM EDTA, pH 7.2) at 37 °C whilst shaking at 200 rpm. Subsequently, the remaining tissue was digested in 30 ml Collagenase solution RPMI 1640 supplemented with 1 mM MgCl2, 1 mM CaCl2, 5% FBS, and 100 units/ml collagenase D (Gibco, Thermo Fisher Scientific) for 1 hr, followed by a Percoll (Sigma) gradient centrifugation at 2000 rpm for 30 min at room temperature. Cells from the gradient were collected, washed and resuspended in PBS supplemented with 2% FCS. Cells were further stained with Fixable Viability Dye eFluor 506 (Thermo Fisher Scientific) and appropriate antibodies and subjected to flow cytometric analysis. For intracellular cytokine, detection cells were stimulated with 25 ng/ml PMA and 750 ng/ml ionomycin (both Sigma-Aldrich) in the presence of GolgiStop (BD Biosciences). For histological analysis, swiss rolls (*Moolenbeek and Ruitenberg, 1981*) were prepared from colons of diseased mice as previously described (*Andersen et al., 2019*).

## Histology

Fixed tissue samples were processed with a tissue processor (Thermo Fisher Scientific). For hematoxylin and eosin (H&E) stainings, histologic evaluation was performed on 5 µm thick sections and stained with hematoxylin and eosin. High power field images (i.e. 400 x magnification) were collected from each colon tissue. At least 4 loci were examined from each slide to ensure unbiased analysis.

## Generation of bone marrow chimeric mice

Mixed bone marrow chimeric mice were generated as previously described (*Hassan et al., 2011*). After six weeks, reconstituted mice were sacrificed and organs were collected as described above. Cells were further stained with Fixable Viability Dye eFluor 506 (Thermo Fisher Scientific) and appropriate antibodies and subjected to flow cytometric analysis.

## Immunization studies

WT and NCOR1-cKO mice were injected via the footpad with 10 µg NP-KLH (Keyhole Limpet Hemocyanin) (Biosearch Technologies) together with 10 µl Imject Alum Adjuvant (Thermo Fisher Scientific). 6 days later, the popliteal (draining) lymph node was isolated and single cell suspension was prepared as described above. Cells were further stained with Fixable Viability Dye eFluor 506 (Thermo Fisher Scientific) and appropriate antibodies and subjected to flow cytometric analysis.

## Low input RNA sequencing of Treg cells

CD4$^+$ T cells were isolated from spleen and LNs as described above and between $5 \times 10^4$ and $5 \times 10^5$ CD4$^+$eGFP$^+$CD44$^{hi}$CD62L$^-$ (effector Treg) and CD4$^+$eGFP$^+$CD44$^{lo}$CD62L$^+$ (naïve Treg) cells from either WT.DEREG or NCOR1-cKO.DEREG mice were FACS purified. Total RNA was prepared from cell lysates using the RNeasy Mini Kit (Qiagen) and RNase-Free DNase Set (Qiagen) according to the manufacturer's protocol. Three biological replicates were generated for each genotype and Treg subset. Each biological replicate was prepared with pooled cells isolated from three individual mice. RNA and library concentrations were determined using Qbit 2.0 Fluorometric Quantitation (Life Technologies). RNA and library integrities were determined using Experion Automated Electrophoresis System (Bio-Rad). Library preparation and RNA Sequencing were performed by the Biomedical Sequencing facility at CeMM (Research Center for Molecular Medicine of the Austrian Academy of Sciences, Vienna, Austria) using low-input Smart-seq2 (*Picelli et al., 2014*). The libraries were sequenced using the Illumina HiSeq 3000 platform and the 50 bp single-read configuration.

## Bioinformatic analysis of RNA sequencing data

Raw sequencing data were processed with Illumina2 bam-tools 1.17 to generate sample-specific, unaligned BAM files. Sequence reads were mapped onto the mouse genome assembly build mm10 (a flavor of GRCm38) using TopHat 2.0.13 (*Saravia et al., 2020*). Gene expression values (reads per kilobase exon per million mapped reads) were calculated with Cufflinks 2.2.1 (*Angelin et al., 2017*).

Volcano plots were generated using the R package EnhancedVolcano (1.22.0). Downstream pathway analysis was performed using the gene set enrichment analysis (GSEA) tools provided by the Broad Institute (*Sanders et al., 1988*) or ingenuity pathway analysis (IPA) (QIAGEN Inc) (*Zhuang et al., 2018*) using default settings. GSEA for hallmark signatures of the MSigDb was performed in R (version 4.3.1) employing the packages msigdbr (7.5.1), fgsea (1.26.0), dplyr (1.1.2), ggplot2(3.4.2), tibble(3.2.1), genekitr(1.2.5), forcats (1.0.0), and org.Mm.eg.db (3.17.0). Differential expression modeling was done with the Bioconductor (3.12) package DESeq2 (1.30.0), running under R 4.0.0.

## iTreg differentiation culture

Naïve CD4$^+$ T cells were isolated from WT.DEREG and NCOR1-cKO.DEREG mice (8–12 weeks) as described above and cultured in a 48-well plate (500.000 cells/well) (Sarstedt) in T cell medium supplemented with 2 ng/ml TGFβ (R&D) and 100 U/ml rhIL-2 (PeproTech) in the presence of plate-bound anti-CD3ε (1 µg/mL) (BD Biosciences) and anti-CD28 (3 µg/mL) (BD Biosciences). After 72 hr, cells were harvested and stained with Fixable Viability Dye eFluor 506 (Thermo Fisher Scientific) and appropriate antibodies and subjected to flow cytometric analysis.

## Treg cell viability assay

WT and NCOR1-cKO splenocytes were cultured in the presence of anti-CD3/anti-CD28 for 72 hr. Cell viability using Fixable Viability Dye eFluor 780 (Thermo Fisher Scientific) was assessed at 0 hr (i.e. ex vivo), 4 hr, or 24 hr of activation.

## Pharmacological activation of liver X receptor (LXR) using GW3965

iTreg cultures were performed as described and simultaneously treated with different concentrations of GW3965 (Sigma) diluted in DMSO (Sigma), as indicated. After 72 hr, cells were harvested and

stained with Fixable Viability Dye eFluor 506 (Thermo Fisher Scientific) and appropriate antibodies and subjected to flow cytometric analysis.

## Filipin III staining

Splenocytes were stained with anti-CD4 and anti-CD25 antibodies for 30 min in PBS supplemented with 2% FCS. After washing, cells were fixed with 2% formaldehyde in PBS for 1 hr at room temperature. After fixation, cells were washed three times, and formaldehyde carryover was quenched using 1.5 mg/ ml Glycine in PBS for 10 min at room temperature. After washing, cells were stained with Filipin III solution (50 μg/ml) (Sigma) in PBS and incubated for 2 hr at room temperature. After staining, cells were washed three times and subjected to flow cytometric analysis.

## CRISPR-Cas9-mediated knockout of NCOR1

All functional assays were performed in IMDM (Gibco, Thermo Fisher Scientific) supplemented with 10% of fetal calf serum (Gibco) and 10 μg/mL of gentamicin (Gibco). Peripheral blood draws were performed from healthy human volunteers in accordance with the Ethics Committee of the Medical University of Vienna (EC number EK 1150/2015). Mononuclear cells were isolated by standard Ficoll-Paque centrifugation. Naïve human CD4+ T cells were isolated using the EasySep Human Naïve CD4 T Cell Iso Kit II (Stem Cell Technologies) according to the manufacturers' instructions. Purity of isolated cells was assessed by flow cytometry and found >95% for all specimens. Subsequently, CRISPR-Cas9 knockout of human NCOR1 was performed as described previously (*Seki and Rutz, 2018*). In detail, 1 μL of a mixture of three NCOR1 specific crRNAs (Alt-R CRISPR-Cas9 crRNA; total concentration 320 μM; sequences:5'-GGA ATC GAA GCG ACC ACG TCT GG-3', 5'-TAA CCA GCC ATC AGA TAC CAA GG-3', 5'-CGG TGT TTC TGC TCC ACA GGA GG-3'; underlined is the PAM sequence) were mixed with 1 μL tracr RNA (320 μM; all Integrated DNA Technologies, Newark, NJ, USA) and hybridized for 10 min at room temperature. The crRNA-tracr RNA duplex was subsequently complexed with 0.7 μL recombinant Cas9 (Alt-R S.p. Cas9 Nuclease V; 10 μg/μL; IDT) for 30 min at 37 °C. Similarly, a control RNP complex was assembled using a non-targeting crRNA (Alt-R CRISPR-Cas9 Negative Control crRNA #1; IDT). For electroporation, 1x10^6 purified naive T cells were resuspended in 15 μL buffer P3 +5 μL Supplement 1 (P3 Primary Cell 4D-NucleofectorTM X Kit S; Lonza) and mixed with 2.7 μL RNP in the 16-well strips provided in the kit. Cells were electroporated on a 4D nucleofector (4D-Nucleofector Core Unit, 4D-Nucleofector X Unit; Lonza) using the pulse code EH100. Immediately afterward, 80 μl pre-warmed fresh medium was added to the cells. After a 1 hr resting period cells were transferred to 24-well plates and incubated for three days in a medium containing 10 U/ mL recombinant human IL-2 (Peprotech) to allow the establishment of the knockout. CRISPR-Cas9 knockout efficiency was determined by Sanger sequencing of the target sites of the three gRNAs and analyzed using the Synthego inference of CRISPR edits analysis tool (ICE v2 CRISPR Analysis Tool; Synthego, Menlo Park, CA). The knockout score defining frameshift insertions/deletions was found to be >65% for at least two of the three loci in all samples tested.

For polarization of induced regulatory T cells (iTreg), cells were preincubated for 1 hr with 100 U/ mL IL-2 (Peprotech) +100 nM all trans-retinoic acid (atRA, Sigma-Aldrich) +5 ng/mL TGFβ (PeproTech), stimulated with anti-CD3/anti-CD28 coated Dynabeads (Thermo Fisher Scientific) (beads:cell ratio 1:2) and cultured for 7 days as described previously (*Gerner et al., 2020*). After polarization, in vitro cultured cells were washed and resuspended in PBS supplemented with 2% FCS. Cells were further stained with appropriate antibodies and subjected to flow cytometric analysis. Intracellular staining for transcription factors was performed using the Foxp3 Staining Buffer Set (Thermo Fisher Scientific) according to the manufacturer's protocol.

## Statistical analysis

No statistical methods were used to predetermine the sample size. All statistical analyses were performed using Prism 8 Software (GraphPad Inc). As indicated in each figure legend, p-values were calculated using either unpaired two-tailed Student's t-test, one-way or two-way ANOVA followed by Tukey's multiple comparison test or a t-test and Wilcoxon-ranked column statistics. No data were excluded and no specific randomization of animals or blinding of investigators was applied.

## Acknowledgements

We thank Johan Auwerx for providing floxed-*Ncor1* mutant mice. WE, CB, and NB were supported by the Austrian Science Fund (FWF) Special Research Program F70. WE was supported by FWF projects P19930, P23641, P26193, P29790, P35372; and by the FWF and Medical University of Vienna doctoral programs (DK W1212) 'Inflammation and Immunity' and (DOC 32 doc.fund) 'TissueHome.' VS. was supported by the FWF and Medical University of Vienna doctoral programs (DK W1212) 'Inflammation and Immunity.' PH was supported by a DOC fellowship of the Austrian Academy of Sciences. DH received a L'Oréal Austria Fellowship supported by the Austrian Commission for UNESCO in cooperation with the Austrian Academy of Sciences. MT was supported by the FWF and Medical University of Vienna doctoral programs (DK W1212) 'Inflammation and Immunity.' AB was supported by the FWF and Medical University of Vienna doctoral programs (DK W1212) 'Inflammation and Immunity.' KS was supported by the FWF project P34728. NB was supported by the FWF project P30885. SK received support via the FWF Special Research Programs F54 and F61. We thank Lois Cavanagh for the English language editing.

## Additional information

### Funding

| Funder | Grant reference number | Author |
|---|---|---|
| Austrian Science Fund | F70 W1212 DOC32 doc. fund | Wilfried Ellmeier |
| Austrian Science Fund | W1212 | Michael Trauner Andreas Bergthaler |
| Austrian Academy of Sciences | Doc fellowship | Patricia Hamminger |
| Austrian Science Fund | 10.55776/P34728 | Klaus Schmetterer |
| Austrian Science Fund | 10.55776/P30885 | Nicole Boucheron |
| Austrian Academy of Sciences | L'Oréal Austria Fellowship | Daniela Hainberger |
| Austrian Science Fund | 10.55776/P29790 | Wilfried Ellmeier |
| Austrian Science Fund | 10.55776/P26193 | Wilfried Ellmeier |
| Austrian Science Fund | 10.55776/P35372 | Wilfried Ellmeier |

The funders had no role in study design, data collection and interpretation, or the decision to submit the work for publication.

### Author contributions

Valentina Stolz, Rafael de Freitas e Silva, Conceptualization, Investigation, Writing - original draft, Writing - review and editing; Ramona Rica, Ci Zhu, Teresa Preglej, Patricia Hamminger, Daniela Hainberger, Marlis Alteneder, Lena Müller, Darina Waltenberger, Anastasiya Hladik, Investigation; Monika Waldherr, Software, Investigation, Writing - review and editing; Benedikt Agerer, Tobias Frey, Thomas Krausgruber, Sylvia Knapp, Klaus Schmetterer, Michael Trauner, Andreas Bergthaler, Investigation, Methodology; Michael Schuster, Software, Methodology; Clarissa Campbell, Christoph Bock, Investigation, Methodology, Writing - review and editing; Nicole Boucheron, Conceptualization, Investigation, Writing - review and editing; Wilfried Ellmeier, Conceptualization, Data curation, Supervision, Funding acquisition, Investigation, Methodology, Writing - original draft, Project administration, Writing - review and editing

### Author ORCIDs

Ramona Rica https://orcid.org/0000-0001-5501-8513
Tobias Frey https://orcid.org/0000-0002-6274-9864
Thomas Krausgruber https://orcid.org/0000-0002-1374-0329
Sylvia Knapp https://orcid.org/0000-0001-9016-5244
Klaus Schmetterer https://orcid.org/0000-0001-9328-4871

Christoph Bock (ID) https://orcid.org/0000-0001-6091-3088
Wilfried Ellmeier (ID) https://orcid.org/0000-0001-8192-8481

## Ethics

Animal experiments were evaluated by the ethics committees of the Medical University of Vienna and approved by the Austrian Federal Ministry for Education, Science and Research (GZ:BMBWF-66.009/0039-V/3b/2019, GZ:BMBWF-66.009/0326-V/3b/2019). Animals were maintained in research facilities of the Department for Biomedical Research at the Medical University of Vienna. Animal husbandry and experiments were performed under national laws in agreement with guidelines of the Federation of European Laboratory Animal Science Associations (FELASA), which correspond to Animal Research: Reporting of in vivo Experiments from the National Center for the Replacement, Refinement and Reduction of Animals in Research (ARRIVE) guidelines.

## Decision letter and Author response

Decision letter https://doi.org/10.7554/eLife.78738.sa1
Author response https://doi.org/10.7554/eLife.78738.sa2

## Additional files

### Supplementary files

• Supplementary file 1. Bioinformatic analysis tables of WT and NCOR1-cKO Treg cells. (**a**) List of differentially expressed genes (DEG) between NCOR1-cKO and WT naive Treg cells. FDR <0.05. (**b**) DEG between NCOR1-cKO and WT effector Treg cells. FDR <0.05. (**c**). Gene Set Enrichment Analysis (GSEA) hallmark gene sets enriched between NCOR1-cKO and WT naïve Treg cells. (**d**) GSEA hallmark gene sets enriched between NCOR1-cKO and WT effector Treg cells. (**e**) Top 50 Canonical Pathways (identified by Ingenuity Pathways Analysis) between NCOR1-cKO and WT naive Treg cells. (**f**) Top 50 Canonical Pathways (identified by Ingenuity Pathways Analysis) between NCOR1-cKO and WT effector Treg cells. (**g**) Naive and effector Treg cell gene sets. The lists show the 100 most DEG between naive and effector WT Treg cells.

• MDAR checklist

### Data availability

All data supporting the findings of this study are available within the article and its supplementary information files. RNA Sequencing data have been deposited in the GEO database under the accession number GSE185984.

The following dataset was generated:

| Author(s) | Year | Dataset title | Dataset URL | Database and Identifier |
|---|---|---|---|---|
| Ellmeier W, Stolz V | 2024 | NCOR1-cKO and WT RNAseq data | https://www.ncbi.nlm.nih.gov/geo/query/acc.cgi?acc=GSE185984 | NCBI Gene Expression Omnibus, GSE185984 |

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
