## [Editor Report]

This study presents important findings on the role of the transcriptional adaptor protein NCOR1 for mouse and human regulatory T (Treg) cell differentiation. The study shows that the LXRbeta – NCOR1 axis restricts the terminal differentiation of Treg cells into effector Tregs. It also shows that, in addition to an impact on effector Treg differentiation, loss of NCOR1 leads to impaired suppressive function of Treg cells. The results are convincing and will contribute to our understanding of Treg cell differentiation and function.

---

## [Decision Letter]

**Decision letter after peer review:**

Thank you for submitting your article "Nuclear receptor corepressor 1 controls regulatory T cell subset differentiation and effector function" for consideration by *eLife*. Your article has been reviewed by 2 peer reviewers, one of whom is a member of our Board of Reviewing Editors, and the evaluation has been overseen by Tadatsugu Taniguchi as the Senior Editor. The reviewers have opted to remain anonymous.

Essential revisions (for the authors):

1) While this manuscript provides various results from animal studies, it lacks basic information about NCOR1-deficiency in Tregs. Based on the previous reports, NCOR1-deficiency might affect cell survival, proliferation, differentiation, phenotype, and function of CD4^+^ T cells including Tregs. The authors should examine possible effects of NCOR1-deficiency on Tregs. For example, in vitro Treg suppression assay, their in vitro proliferation/cell death after CD3/CD28 stimulation, Treg marker gene expression, their cytokine expression, and their possible effects on interacting APCs, should be examined with NCOR1-deficient Tregs. In particular, since TCR stimulation is a key factor for driving T cells from naive to effector, TCR-mediated changes including signaling pathways, phenotypical changes, cell proliferation, and transcriptional regulations, should be addressed with NCOR1-deficient Tregs.

2) The authors claim that the LXR-NCOR1 axis regulates effector Treg differentiation. Can Foxp3+ T cells be generated under the NCOR1-deficient condition? In the colitis model (Figure 6), NCOR1-deficient Tregs did not show Treg suppressive function, and also CD4^+^ T cells including Tregs were dramatically reduced under NCOR1-deficiency (Figure 1). The results suggest that NCOR1 can be concerned with maintaining T cell function, rather than Treg conversion from naive to effector, and that NCOR1-deficiency does not exhibit a specific effect on Tregs.

3) To claim the role of the LXR-NCOR1 axis in effector Treg differentiation, the authors need to address the following issues. Does LXR deletion cancel the effects? Does LXR and NCOR1 directly bind to the promoter/enhancer of MYC? Does T cell activation break the interaction of LXR and NCOR1? Does the phenotype of NCOR1 loss show a correlation with the phenotype of Treg activation? How about LXR loss?

4) Tfr cells are presented as one example of effector Treg cells. However, there is evidence that Tfr cells themselves can be found as immature (the ones in circulation are notably immature), and at different stages of maturation (the most mature Tfr cells lose CD25 expression). The different impact of IL-2 on the maturation of effector Treg cells and Tfr cells (the differentiation is suppressed by IL-2) also makes this issue less clear. This possibility regarding Tfr cells needs to be discussed.

*Reviewer #1 (Recommendations for the authors):*

1. While this manuscript provides various results from animal studies, it lacks basic information about NCOR1-deficiency in Tregs. Based on the previous reports, NCOR1-deficiency might affect cell survival, proliferation, differentiation, phenotype, and function of CD4^+^ T cells including Tregs. The authors should examine possible effects of NCOR1-deficiency on Tregs. For example, in vitro Treg suppression assay, in vitro proliferation/cell death analysis after CD3/CD28 stimulation, Treg marker gene expression, and cytokine expression, APC interaction/activation, should be examined with NCOR1-deficient Tregs. In particular, since TCR stimulation is a key factor for driving T cells from naive to effector, TCR-mediated changes including signaling pathways, phenotypical changes, cell proliferation, and transcriptional regulations, should be addressed with NCOR1-deficient Tregs.

2. Based on the expression of CD44 and CD62L, the authors claim that NCOR1 deficiency enhances a conversion from naive to effector in Tregs. The development of functionally mature effector Treg cells should not be simply defined by the expression of two markers. Does NCOR1-deficiency induce effector-type epigenetic status, cytokine secretion, cell growth, cell morphology, and especially Treg suppressive function?

3. The authors claim that the LXR-NCOR1 axis regulates effector Treg differentiation. Can Foxp3+ T cells be generated under the NCOR1-deficient condition? In the colitis model (Figure 6), NCOR1-deficient Tregs did not show Treg suppressive function, and also CD4^+^ T cells including Tregs were dramatically reduced under NCOR1-deficiency (Figure 1). The results suggest that NCOR1 can be concerned with maintaining T cell function, rather than Treg conversion from naive to effector, and that NCOR1-deficiency does not exhibit a specific effect on Tregs.

4. Although the authors claim that LXR-NCOR1 axis regulates effector Treg differentiation, the results presented here are insufficient for their claim. To claim the role of the LXR-NCOR1 axis in Treg differentiation, the authors need to address the following issues. Does LXR deletion cancel the effects? Does LXR and NCOR1 directly bind to the promoter/enhancer of MYC? Does T cell activation break the interaction of LXR and NCOR1? Does the phenotype of NCOR1 loss show a correlation with the phenotype of Treg activation? How about LXR loss?

5. Furthermore, NCOR1 has been shown to interact with a variety of nuclear receptors. Have the authors checked transcriptional control by other nuclear receptors that can bind to NCOR1 in NCOR1-deficient Tregs?

*Reviewer #2 (Recommendations for the authors):*

The manuscript describes a well-designed set of experiments that are very conclusive in supporting the authors' conclusions. In particular, the major findings are convincingly demonstrated by the experimental results. There are, however, two issues that still require the authors' attention.

1. Tfr cells are presented as one example of eTreg cells. However, there is evidence that Tfr cells themselves can be found as immature (the ones in circulation are notably immature), and at different stages of maturation (the most mature Tfr cells lose CD25 expression). The impact different impact of IL-2 on the maturation of eTreg cells and Tfr cells (the differentiation is suppressed by IL-2) also makes this issue less clear. It is likely that the authors' assessment takes into consideration the mature tissue CD25+ Tfr cells. However, this issue should be discussed with greater attention to the different maturation states of Tfr cells.

2. The evidence supporting the role of NCOR1 in maintaining the suppressive function of Treg cells is not as strong as the evidence supporting other claims. This claim is a particular concern as one anticipated that eTreg cells would have a greater suppressive function than non-effector Treg cells. The fact that NCOR1 deficiency in Treg cells prevented their ability to protect mice from colitis may be due to other reasons beyond the suppressive function alone. A direct assessment of the suppressive function of NCOR1-deficient Treg cells under in vitro assays could provide conclusive evidence regarding this claim.

---

## [Author Response]

Essential revisions (for the authors):1) While this manuscript provides various results from animal studies, it lacks basic information about NCOR1-deficiency in Tregs. Based on the previous reports, NCOR1-deficiency might affect cell survival, proliferation, differentiation, phenotype, and function of CD4^+^ T cells including Tregs. The authors should examine possible effects of NCOR1-deficiency on Tregs. For example, in vitro Treg suppression assay, their in vitro proliferation/cell death after CD3/CD28 stimulation, Treg marker gene expression, their cytokine expression, and their possible effects on interacting APCs, should be examined with NCOR1-deficient Tregs. In particular, since TCR stimulation is a key factor for driving T cells from naive to effector, TCR-mediated changes including signaling pathways, phenotypical changes, cell proliferation, and transcriptional regulations, should be addressed with NCOR1-deficient Tregs.

As suggested by the Reviewers and recommended by the Editor, we have performed the required/suggested experiments to better characterize NCOR1-deficient Treg cells. We describe the different types of experiments in detail on the next pages in this point-to-point reply to the individual Reviewer’s comments. The new data have been inserted throughout the revised manuscript. A few new data have been included into the main figures, while the majority of the new data were added as supplementary figure panels.

The following data have been added to address the essential revisions described in comment #1:

Cytokine expression: Figure 4—figure supplement 1a and 1b

Suppressive function: Figure 4—figure supplement 1c and 1d

Effects on interacting APCs: Data only shown in point-to-point reply

Proliferation and survival: Figure 1—figure supplement 2a, 2b and 2c

Treg cell marker gene expression: Figure 1—figure supplement 1c and 1d and figure shown in pointto-point reply.

Signaling in Treg cells: Figure 1—figure supplement 2h

Treg cell viability: Figure 1—figure supplement 2c

FoxP3 stability: Figure 2c

Thymic Treg cell differentiation: Figure 1—figure supplement 2e, 2f and 2g. CD25 expression on Treg cells: Figure 1h and 1i.

2) The authors claim that the LXR-NCOR1 axis regulates effector Treg differentiation. Can Foxp3+ T cells be generated under the NCOR1-deficient condition? In the colitis model (Figure 6), NCOR1-deficient Tregs did not show Treg suppressive function, and also CD4^+^ T cells including Tregs were dramatically reduced under NCOR1-deficiency (Figure 1). The results suggest that NCOR1 can be concerned with maintaining T cell function, rather than Treg conversion from naive to effector, and that NCOR1-deficiency does not exhibit a specific effect on Tregs.

We agree with the Editor and Reviewer that NCOR1 is an important factor for T cells in general and that its function is not restricted to Treg cells. We previously showed that NCOR1 is essential for positive selection as well as for setting up the transcriptional landscape in Th1 and Th17 cells. It was not our intention to convey the view that NCOR1 specifically regulates the Treg cell lineage only. However, our data clearly demonstrate that NCOR1 also has an essential function in the Treg cell lineage, which in part is different to its function in other Th lineages. We have addressed this comment by including this point in the Discussion section (page 20).

3) To claim the role of the LXR-NCOR1 axis in effector Treg differentiation, the authors need to address the following issues. Does LXR deletion cancel the effects? Does LXR and NCOR1 directly bind to the promoter/enhancer of MYC? Does T cell activation break the interaction of LXR and NCOR1? Does the phenotype of NCOR1 loss show a correlation with the phenotype of Treg activation? How about LXR loss?

We agree that it is important to investigate the link between LXRb and NCOR1 in more detail. We also are thankful for this comment, since the experiments in response to this comment provided a better and more detailed insight whether the phenotype observed in NCOR1-cKO Tregs is linked with LXRß. For the revised manuscript we decided to use a genetic approach and generated LXRb single and NCOR1/LXRb double knockout mouse mice. It took us some time and efforts to generate this data, since we had to import LXRb knockout animals into our animal facility via embryo transfer followed by intercrossing with NCOR1-cKO mice. We describe our results in detail on page 11 of this reply letter and we inserted a new sub-chapter (page 13) as well as a new main figure 7 (and a new Figure 7—figure supplement 2 and Figure 7—figure supplement 3a) in our revised manuscript. The generation of the new mouse lines resulted in three novel findings:

Loss of LXRb did not revert alterations in naïve to effector Treg cell frequencies induced by NCOR1 deletion. Based on these data we propose that NCOR1 controls naïve and effector Treg cell state and subset composition in an LXRb-independent manner.Intracellular MYC staining revealed that MYC protein level were significantly elevated in naïve LXRb-cKO and NCOR1-LXRb-cDKO FOXP3^+^ Treg cells compared to WT Treg cells. In contrast, NCOR1-cKO Treg cells (after including additional samples in the analysis) did not show an upregulation of MYC protein despite an upregulation of MYC target genes. However, elevated MYC protein levels in LXRß-deficient Treg cells were not sufficient to induce an increase in effector Treg cells. This suggest that NCOR1 might restrain MYC activity, which might contribute to the observed increase in effector Treg cells upon NCOR1 deletion.The generation of DKO mice also provided novel insight into the generation of Treg cells. The Treg cell frequency within the LXRb-cKO CD4^+^ T cell population in the spleen was higher compared to WT littermate controls (as also previously reported in PMID: 33373442), while FOXP3^+^ T cells within the CD4^+^ T cell lineage were reduced in the absence of NCOR1. Since LXRb deletion on top of NCOR1 deletion reverted the relative decrease in Treg cells, this suggests that NCOR1 controls the generation of Treg cells in an LXRb-dependent manner.

We therefore propose a model that NCOR1 suppresses LXRb or counteracts LXRb-induced pathways, which itself restrains the generation of FOXP3^+^ Treg cells. However, NCOR1 maintains naïve and effector cell states independently of LXRß. This model is shown as graphical abstract and as Figure 7—figure supplement 4. See also page 19 (Discussion section) in the revised manuscript.

The following data have been added to address the essential revisions described in comment #13: T cell phenotype of LXRb-cKO and LXRb/NCOR1-double KO mice: Figure 7; Figure 7—figure supplement 2, Figure 1—figure supplement 3a.

We also agree with the Editor (and with Reviewer #1) that it would be very interesting to identify NCOR1 and/or LXRb binding sites at a genome-wide level (also in the absence of either one or the other) and to study whether the interaction between NCOR1 and LXRb is disrupted upon T cell activation. We first focused on assessing the impact of LXRb deletion on top of NCOR1-deficiency using genetic (i.e. gene targeting) approaches. This uncovered that NCOR1-regulates naïve and effector Treg cell states and subset composition in an LXRb-independent manner. Therefore, we did not follow-up a potential interaction between NCOR1 and LXRß. In addition, we also did not have established protocols in the lab to perform LXRb ChIP assays in WT CD4^+^ T cells to determine LXRb binding sites. Moreover, the number of CD4^+^ T cells is severely reduced in NCOR1-cKO and in LXRb-cKO mice, this also would make it technically challenging to identify NCOR1 or LXRb binding sites in the absence of either one or the other. We are very sorry that we therefore were not able to address the issue concerning binding of NCOR1 and LXRb to target sites. This was the only comment that we didn’t address in the revised manuscript. In order to acknowledge the importance of such a dataset, we introduced a short “limitation of our study” paragraph in our discussion where this issue is mentioned (page 20). We hope that the Editor and the Reviewers nevertheless appreciate our efforts in comprehensively revising our manuscript, including the generation of LXRb single and NCOR1/LXRb double knockout mice which provided novel insight into the potential interplay of these two factors.

4) Tfr cells are presented as one example of effector Treg cells. However, there is evidence that Tfr cells themselves can be found as immature (the ones in circulation are notably immature), and at different stages of maturation (the most mature Tfr cells lose CD25 expression). The different impact of IL-2 on the maturation of effector Treg cells and Tfr cells (the differentiation is suppressed by IL-2) also makes this issue less clear. This possibility regarding Tfr cells needs to be discussed.

We have addressed this issued in the revised manuscript. We also provide additional data (please see page 15 in this reply letter for a detailed description) as well as discuss this possibility.

Reviewer #1 (Recommendations for the authors):1. While this manuscript provides various results from animal studies, it lacks basic information about NCOR1-deficiency in Tregs. Based on the previous reports, NCOR1-deficiency might affect cell survival, proliferation, differentiation, phenotype, and function of CD4^+^ T cells including Tregs. The authors should examine possible effects of NCOR1-deficiency on Tregs. For example, in vitro Treg suppression assay, in vitro proliferation/cell death analysis after CD3/CD28 stimulation, Treg marker gene expression, and cytokine expression, APC interaction/activation, should be examined with NCOR1-deficient Tregs.2. Based on the expression of CD44 and CD62L, the authors claim that NCOR1 deficiency enhances a conversion from naive to effector in Tregs. The development of functionally mature effector Treg cells should not be simply defined by the expression of two markers. Does NCOR1-deficiency induce effector-type epigenetic status, cytokine secretion, cell growth, cell morphology, and especially Treg suppressive function?

We appreciate the comments raised by this Reviewer #1 and the suggestion for additional experiments. These are all important questions and we also agree with the comment that additional Treg cell features (besides CD44 and CD62L expression) should be included in the characterization. For the revised manuscript we added an in-depth characterization of NCOR1-deficient Treg cells. We performed many additional experiments and provide new data on the following topics:

Cytokine expression: We investigated whether NCOR1 controls the expression of the cytokine TGFb. Ex vivo PMA/ionomycin stimulation of isolated Treg cells revealed an increase in TGFb expression in NCOR1-cKO Treg cells compared to WT Treg cells. These data have been added in the revised manuscript on page 10 and as Figure 4—figure supplement 1a and 1b.

Suppressive function: We performed in vitro suppression assays and tested the proliferation (assessed by Cell trace violet dye) of WT responder CD4^+^ T cells activated by dendritic cells (DC) in the presence of increasing amount of WT and NCOR1-cKO Treg cells. There was no difference in the suppressive activity of splenic NCOR1-deficient FOXP3^+^ Treg cells compared to WT Treg cells. In the revised manuscript, this is described on page 10 and added as Figure 4—figure supplement 1c and 1d.

Effects on interacting APCs: Treg cells can control the co-stimulatory function of DCs by modulating CD80/CD86 expression, as nicely shown e.g. in PMID:18635688. In this publication, the authors used DO11.10 TCR transgenic Treg cells and co-cultured DC and Treg cells in the presence of OVA. Since we didn’t have NCOR1-cKO mice on a TCR tg background, we tried to adapt the assay and used anti-CD3-stimulation (instead of OVA) and compared the impact of WT and NCOR1-cKO Treg cells on co-cultured DCs in vitro*.* There was no difference in the expression of CD80, CD86 and MHCII on CD11c^+^ DCs co-cultured with WT or NCOR1-cKO Treg cells after 24h. In fact, the presence of WT Treg cells had no impact on CD80, CD86 and MHCII expression on DCs, indicating that the assay didn’t work with anti-CD3 stimulation. We therefore decided to focus our revision experiments on the other experiments described in this point-to-point reply.

**Author response image 1. sa2fig1:** (**a**) Flow cytometry analysis of CD4 and CD11c expression in co-cultures of splenic dendritic cells (DCs) with either WT of NCOR1-cKO Treg cells after 24 hours. DC and Treg cells were cocultured in the presence of anti-CD3. (**b**) Flow cytometry analysis of CD80, CD86 and MHCII expression in DCs from co-cultures as described in (a). The 1^st^ row displays DC only cultures, while rows 2-6 and rows 7-11 show DC co-culture with WT and NCOR1-cko Treg cells, respectively. Each row represents an independent biological sample of Treg cells.

Proliferation and survival: We also analyzed whether alterations in the proliferation and/or survival of NCOR1-cKO Treg cells contribute to the relative reduction of Treg cells within the peripheral CD4^+^ T cell population. The percentage of Treg cells expressing Ki67, a marker associated with cell proliferation, as well as the expression levels of Ki67 (gMFI) in FOXP3^+^ cells was similar between ex vivo analyzed splenic WT and NCOR1-cKO Treg cells (shown in the revised manuscript on page 5 and in Figure 1—figure supplement 2a and 2b). Similarly, there was also no difference in cell viability between ex vivo analyzed WT and NCOR1-cKO Treg cells (timepoint 0h; Figure 1—figure supplement 2c). These data suggest that Treg cell proliferation and survival at steady state are not controlled by

NCOR1.

Treg cell marker gene expression: We hope that we understood the comment concerning Treg cell marker gene expression in the way meant by Reviewer #1. A comprehensive ex vivo analysis of marker expression (KLRG1, CD69, CD25, ICOS and GITR) on Treg cells was already included in the initially submitted manuscript. However, for the revised version, we included CTLA-4 expression data. Although there was a slight tendency that CTLA-4 expression (MFI) is reduced in NCOR1-cKO Treg cells isolated from the spleen and LNs compared to their WT counterparts, the difference did not reach statistical significance. The data were added as a panel in Figure 1—figure supplement 1c and 1d and are mentioned in the text on page 5.

In case Reviewer #1 suggested that we should determine marker mRNA expression (instead of protein expression by flow cytometry) in Treg cells, we re-analyzed our RNA-seq data of naïve and effector WT and NCOR1-cKO Treg cells and made a graph showing the expression of the abovementioned marker genes. *Ctla4*, *Il2ra* and *Tnfrsf18* were down-regulated in splenic naïve NCOR1cKO Treg cells, while there was no difference in *Klrg1*, *Cd69*, *Icos*, *Cd44* and *Sell*. In splenic effector Treg cells, there was no difference in the expression levels of these genes. Therefore, differences observed with respect to Treg cell marker expression as show in Figure 1—figure supplement 1 were rather due to changes in the percentages of naïve and effect Treg cell subsets and not due to gene expression differences in the respective subsets. We didn’t add the gene expression data in the revised manuscript, since we already have included many additional figure panels. However, we display them in the point-to-point reply as a Author response image 2. If Reviewer #1 thinks that it would be better to show them in the manuscript, then we are more than happy to include them.

**Author response image 2. sa2fig2:** Summary showing the expression levels (values showing as fragments per kilobase of transcript per million mapped reads; FPKM) of the indicated genes in naive and effector WT and NCOR1-cKO Treg cells as determined by RNA-seq. Each bar represents 3 mice per group. Mean ±}SD is shown. *P <0.05, **P < 0.01, and ***P < 0.001 (unpaired 2-tailed Student’s t test).

In particular, since TCR stimulation is a key factor for driving T cells from naive to effector, TCR-mediated changes including signaling pathways, phenotypical changes, cell proliferation, and transcriptional regulations, should be addressed with NCOR1-deficient Tregs.

To study whether NCOR1 controls Treg cell activation pathways in response to TCR triggering and thus to enhanced effector differentiation, we analyzed TCR proximal signaling pathways in ex vivo WT and NCOR1-cKO Treg cells. We assessed the expression of phospho-ERK ^T202/Y204^, phospho-AKT^S473^ and phospho-S6^S240/244^ in FOXP3^+^ Treg cells cultured in the presence of anti-CD3/anti-CD28 after 10 minutes, 2 hours and 24 hours. Despite a trend for higher levels of phospho-ERK^T202/Y204^, phospho-AKT^S473^ and phospho-S6^S240/244^ in NCOR1-cKO Treg cells at several early time points, we only observed a significant difference for phospho-AKT^S473^ at the 2 hours timepoint, which might indicate altered mTORC2 activation ^29, 30^. Overall, these data suggest that NCOR1-cKO Treg cells might display a slightly higher transient degree of activation after TCR triggering. These data are briefly discussed in the revised manuscript on page 7 and have been added as Figure 1—figure supplement 2h.

Treg cell viability: In response to the comments of Reviewer #1, we also assessed Treg cell viability upon activation. For this, we isolated WT and NCOR1-cKO splenocytes and cultured them in the presence of anti-CD3/anti-CD28 over a time period of 72 hours. While there was no significant difference in the viability between WT and NCOR1-cKO Treg cells after 0, 4 or 24 hours of activation, we observed a significant decrease in the percentage of viable NCOR1-cKO Treg cells after 72 hours. In the revised manuscript we added this data in Figure 1—figure supplement 2c and mention the data on page 7.

FoxP3 stability: It was shown that iTreg cells downregulate FOXP3 expression upon restimulation with anti-CD3/anti-CD28 in the absence of TGFb (PMID: 17298177). To test whether NCOR1 is important for the maintenance of FOXP3 expression, we first generated WT and NCOR1-cKO iTreg cells and restimulated the cells with anti-CD3/anti-CD28 for 4 days. Although we observed a decrease in the percentage of FOXP3-expressing cells during the 4-day restimulation culture period, there was no difference in the percentage of FOXP3^+^ cells between WT and NCOR1-cKO Treg cells, indicating that NCOR1 function is dispensable for the physiological maintenance of FOXP3 expression in iTreg cells. These data have been added as Figure 2c and briefly mentioned on page 8.

3. The authors claim that the LXR-NCOR1 axis regulates effector Treg differentiation. Can Foxp3+ T cells be generated under the NCOR1-deficient condition?

We hope that we correctly understood the point raised. FOXP3^+^ cells can be generated in the absence of NCOR1, both in vivo as well as in vitro from naïve CD4^+^ T cells, although at reduced frequencies. For the revised manuscript, we analyzed Treg cell differentiation in the thymus in more detail. Thymic Treg cells develop from CD4SP CD25^+^ progenitor cells which start to express FOXP3 (PMIDs: 18199417, 18199418) although FOXP3 can also be induced in CD4SP cells before CD25 expression (PMID: 23746651). Both CD25^+^FOXP3^+^ and CD25^–^FOXP3^+^ subsets were reduced within the NCOR1-cKO CD4SP population in comparison to the corresponding WT CD4SP population. However, the fraction of thymic CD44^hi^CD62L^–^ NCOR1-cKO Treg cells was increased (both within the CD25^–^ and CD25^+^ subsets). These data indicate that the relative reduction of Treg cells within the CD4^+^ T cell population as well as the change in the relative abundance of naïve and effector Treg cells in NCOR1-cKO mice is (in part) already established during Treg cellgeneration in the thymus. We have added these data in a new Figure 1—figure supplement 2e, 2f and 2g and describe the data on page 6.

In the colitis model (Figure 6), NCOR1-deficient Tregs did not show Treg suppressive function, and also CD4^+^ T cells including Tregs were dramatically reduced under NCOR1-deficiency (Figure 1). The results suggest that NCOR1 can be concerned with maintaining T cell function, rather than Treg conversion from naive to effector, and that NCOR1-deficiency does not exhibit a specific effect on Tregs.

We agree with the reviewer that NCOR1 function is not restricted to Treg cells. We previously showed that NCOR1 is essential for setting up the transcriptional landscape in Th1 and Th17 cells. It was not our intention to convey the view that NCOR1 specifically regulates the Treg cell lineage only. We added a few sentences in the Discussion section to provide a more balanced view about this topic (page 20).

4. Although the authors claim that LXR-NCOR1 axis regulates effector Treg differentiation, the results presented here are insufficient for their claim. To claim the role of the LXR-NCOR1 axis in Treg differentiation, the authors need to address the following issues. Does LXR deletion cancel the effects? Does LXR and NCOR1 directly bind to the promoter/enhancer of MYC? Does T cell activation break the interaction of LXR and NCOR1? Does the phenotype of NCOR1 loss show a correlation with the phenotype of Treg activation? How about LXR loss?

We also are thankful for this comment, since the experiments in response to this comment provided a better and more detailed insight whether the phenotype observed in NCOR1-cKO Tregs is linked with LXRß. We generated LXRb single- and NCOR1/LXRb double-knockout mice to address (some) of these questions. It took us some to time to generate this data, since we had to import LXRb knockout animals into our animal facility (via embryo transfer). To obtain a sufficiently high number of mice for the analysis, we established independent breeding colonies for NCOR1cKO, LXRb-cKO as well as for NCOR1-LXRb-cDKO mice and used the corresponding *Cre*-negative littermates as WT controls (i.e. *Ncor1*^f/f^, designated as WT^Ncor1^; *Nr1h2*^f/f^, as WT^Lxrb^; and *Ncor1*^f/f^,*Nr1h2*^f/f^, as WT^Ncor1/Lxrb^, respectively). In agreement with a previous study, we observed a strong reduction in the frequencies of splenic T cells in the absence of LXRb (Figure 7—figure supplement 2a and 2b). Within the TCRb^+^ population, CD4^+^ T cells were slightly reduced, while the frequency of CD8^+^ T cells was not altered (Figure 7—figure supplement 2b and 2c). Of note, NCOR1LXRb-cDKO mice also showed a severe reduction of T cell numbers and slightly lower frequencies of CD4^+^ T cells as observed in LXRb-cKO mice (Figure 7—figure supplement 2a, 2b and 2c). Interestingly, the Treg cell frequency within the LXRb-cKO CD4^+^ T cell population in the spleen was higher compared to WT littermate controls (Figure 7a and 7b), in contrast to NCOR1-cKO mice where a 2-fold relative decrease was observed (Figure 7a and 7b). NCOR1-LXRb-cDKO mice also showed a higher representation of Treg cells amongst the CD4^+^ T cell population (Figure 7a and 7b), suggesting that loss of LXRb is sufficient to rescue the low Treg cell frequency phenotype seen in NCOR1 cKO mice. Within the LXRb-cKO Treg cell population, there was a tendency of a mild increase in CD44^+^CD62L^–^ effector cell frequencies (P=0.058), while CD44^lo^CD62L^+^ naïve Treg cells were not changed (Figure 7c and 7d). This suggests that LXRb deletion using CD4-*Cre* only had a small effect on naïve and effector Treg cell composition. In contrast, splenocytes from NCOR1-LXRb-cDKO mice displayed a strong increase in the frequency of CD44^hi^CD62L^–^ effector Treg cells along with a severe decrease in naïve Treg cells (Figure 7c and 7d), similar to NCOR1-cKO mice. This indicates that loss of LXRb did not revert alterations in naïve to effector Treg cell frequencies induced by NCOR1 deletion. Within the Treg cell population of some splenocytes from NCOR1-LXRb-cDKO mice more than 50% of effector Treg cells were detected (Figure 7c and 7d), a frequency not reached in NCOR1cKO mice (Figure 1b and 7d). This might suggest a minor contribution of LXRb in the generation of effector Treg cells, at least in the absence of NCOR1.

We also investigated whether MYC protein expression levels were affected by the absence of NCOR1, LXRb or both. After including additional samples, intracellular staining revealed no difference in MYC protein levels between WT^Ncor1^ and NCOR1-cKO naïve Treg cells and slightly lower MYC levels in effector Treg cells (Figure 7—figure supplement 3a), contrasting with the observed upregulation of *Myc* mRNA levels and MYC target genes. MYC protein expression was significantly increased in naïve CD44^lo^CD62L^+^ LXRb-cKO and NCOR1-LXRb-cDKO FOXP3^+^ Treg cells in comparison to the respective WT^Lxrb^ and WT^Ncor1/Lxrb^ controls (Figure 7—figure supplement 3a), suggesting that loss of LXRb has a dominant effect on MYC levels in this Treg cell subset. However, elevated MYC protein levels in LXRß-deficient Treg cells were not sufficient to induce an increase in effector Treg cells. This suggest that NCOR1 might restrain MYC activity, which might contribute to the observed increase in effector Treg cells upon NCOR1 deletion. Of note, while LXRb-cKO mice also displayed higher MYC protein levels in effector Treg cells relative to their WT counterparts, this was reverted in NCOR1-LXRb-cDKO animals. Taken together, these data indicate that:

NCOR1 controls naïve and effector Treg cell states and the relative distribution of Treg cell subsets in an LXRb-independent manner.Moreover, our data suggest that NCOR1 restrains MYC activity, which might contribute to the observed increase in effector Treg cells upon NCOR1 deletion.The generation of DKO mice also provided novel insight into the generation of Treg cells. The Treg cell frequency within the LXRb-cKO CD4^+^ T cell population in the spleen was higher compared to WT littermate controls (as also previously reported in PMID: 33373442), while FOXP3^+^ T cells within the CD4^+^ T cell lineage were reduced in the absence of NCOR1. Since LXRb deletion on top of NCOR1 deletion reverted the relative decrease in Treg cells, this suggests that NCOR1 controls the generation of Treg cells in an LXRb-dependent manner.

All data have been included in a new Figure 7, in a new Figure 7—figure supplement 2 and new Figure 7—figure supplement 3a panel and are described in the revised manuscript from page 13 on. We also included a model of how NCOR1 regulated Treg cell differentiation and naïve and effector Treg cell states (Figure 7—figure supplement 4).

We also agree with Reviewer #1 that it would be very interesting to identify NCOR1 and/or LXRb binding sites at a genome-wide level (also in the absence of either one or the other) and to study whether the interaction between NCOR1 and LXRb is disrupted upon T cell activation. Our new data revealed that NCOR1-regulates naïve and effector Treg cell states and subset composition in an LXRb-independent manner. Therefore, we did not follow-up a potential interaction between NCOR1 and LXRß. Furthermore, we didn’t have established protocols in the lab to perform LXRb ChIP assays in WT CD4^+^ T cells to determine LXRb binding sites. Moreover, the number of CD4^+^ T cells is severely reduced in NCOR1-cKO and in LXRb-cKO mice, thus making this technically challenging to identify NCOR1 or LXRb binding sites in the absence of either one or the other. We are very sorry that we therefore were not able to address the issue concerning binding of NCOR1-LXRb to target sites. This was the only comment that we didn’t address in the revised manuscript. In order to acknowledge the importance of such a dataset, we introduced a short “limitation of our study” paragraph in our discussion where this issue is mentioned (page 20). We hope that Reviewer #1 nevertheless appreciates our efforts in addressing the role of the interaction between NCOR1 and LXRb by generating LXRb single- and LXRb/NCOR1 double-knockout mice. This resulted in novel insights into the potential interplay of these two factors.

5. Furthermore, NCOR1 has been shown to interact with a variety of nuclear receptors. Have the authors checked transcriptional control by other nuclear receptors that can bind to NCOR1 in NCOR1-deficient Tregs?

Our Ingenuity Pathway Analysis (see Figure 6—figure supplement 1a) indicated in addition to “LXRb/RXR activation” also alterations in “TR/RXR activation”. We therefore re-analyzed the expression of other NCOR1-interacting nuclear receptors in NCOR1-deficient Treg cells compared to WT Treg cells. We detected dysregulated expression of some, but not all nuclear receptors in the absence of NCOR1, including LDLR, PPARg, VDR, REV-ERBa, THRa, and RARa. This data might indicate that other nuclear receptors might be affected by NCOR1 in Treg cells. Since the handling Editor decided that it wasn’t part of the essential revisions, we didn’t follow up whether they display altered transcriptional control. However, we included a sentence in the discussion (page 20) to mention that the activity of other nuclear receptors might be affected by loss of NCOR1.

**Author response image 3. sa2fig3:** Summary showing the expression levels (values showing as fragments per kilobase of transcript per million mapped reads; FPKM) of the indicated genes in naïve and effector WT and NCOR1cKO Treg cells as determined by RNA-seq. Each symbol represents one sample (2 mice per sample, 3 independent sample preparations steps).

Reviewer #2 (Recommendations for the authors):The manuscript describes a well-designed set of experiments that are very conclusive in supporting the authors' conclusions. In particular, the major findings are convincingly demonstrated by the experimental results. There are, however, two issues that still require the authors' attention.1. Tfr cells are presented as one example of eTreg cells. However, there is evidence that Tfr cells themselves can be found as immature (the ones in circulation are notably immature), and at different stages of maturation (the most mature Tfr cells lose CD25 expression). The impact different impact of IL-2 on the maturation of eTreg cells and Tfr cells (the differentiation is suppressed by IL-2) also makes this issue less clear. It is likely that the authors' assessment takes into consideration the mature tissue CD25+ Tfr cells. However, this issue should be discussed with greater attention to the different maturation states of Tfr cells.

This is a great comment referring to the study by Wing et al. (PMID:28698369). We therefore reanalyzed our Tfr data and assessed CD25 expression (CD25 was included in our Ab panel). This revealed that the percentage of CD25^+^ cells within the Tfr population (defined as CD44^hi^PD1^+^CXCR5^+^FOXP3^+^ CD4^+^ T cells) is reduced in NCOR1-cKO mice compared to WT counterparts. Thus, loss of NCOR1 results in an increase in the fraction of the most mature CD25^–^ Tfr cells. This supports our finding that NCOR1 controls naive to effector Treg cell conversion. We have included this data in the revised manuscript on page 6/7 and show them as Figure 1h and 1i.

2. The evidence supporting the role of NCOR1 in maintaining the suppressive function of Treg cells is not as strong as the evidence supporting other claims. This claim is a particular concern as one anticipated that eTreg cells would have a greater suppressive function than non-effector Treg cells. The fact that NCOR1 deficiency in Treg cells prevented their ability to protect mice from colitis may be due to other reasons beyond the suppressive function alone. A direct assessment of the suppressive function of NCOR1-deficient Treg cells under in vitro assays could provide conclusive evidence regarding this claim.

We agree with the Reviewer #2 that it is important to determine the in vitro suppressive activity of NCOR1-deficient Treg cells. This issue was also raised by Reviewer #1. We performed in vitro suppression assays and tested the proliferation (assessed by Cell trace violet dye) of WT responder CD4^+^ T cells activated by dendritic cells (DC) in the presence of increasing amount of WT and NCOR1-cKO Treg cells. There was no difference in the suppressive activity of splenic NCOR1deficient FOXP3^+^ Treg cells compared to WT Treg cells. In the revised manuscript, this is described on page 10 and added as Figure 4—figure supplement 1c and 1d.

As shown in our reply to comment #1 of Reviewer #1 (on page 5), TGFß production was increased. However, this didn’t result in an increase in the suppressive activity of NCOR1-deficient Treg cells in in vitro suppression assays.